# Weight Diffusion for Future: Learn to Generalize in Non-Stationary Environments

**Mixue Xie**
Beijing Institute of Technology
mxxie@bit.edu.cn

**Shuang Li**✉
Beijing Institute of Technology
shuangli@bit.edu.cn

**Binhui Xie**
Beijing Institute of Technology
binhuixie@bit.edu.cn

**Chi Harold Liu**
Beijing Institute of Technology
chiliu@bit.edu.cn

**Jian Liang**
Kuaishou Technology
liangjian03@kuaishou.com

**Zixun Sun**
Tencent
sunzixun@126.com

**Ke Feng**
Tencent
richardfeng@tencent.com

**Chengwei Zhu**
Tencent
chavezzhu@tencent.com

## Abstract

Enabling deep models to generalize in non-stationary environments is vital for real-world machine learning, as data distributions are often found to continually change. Recently, evolving domain generalization (EDG) has emerged to tackle the domain generalization in a time-varying system, where the domain gradually evolves over time in an underlying continuous structure. Nevertheless, it typically assumes multiple source domains simultaneously ready. It still remains an open problem to address EDG in the domain-incremental setting, where source domains are non-static and arrive sequentially to mimic the evolution of training domains. To this end, we propose *Weight Diffusion (W-Diff)*, a novel framework that utilizes the conditional diffusion model in the parameter space to learn the evolving pattern of classifiers during the domain-incremental training process. Specifically, the diffusion model is conditioned on the classifier weights of different historical domain *(regarded as a reference point)* and the prototypes of current domain, to learn the evolution from the reference point to the classifier weights of current domain *(regarded as the anchor point)*. In addition, a domain-shared feature encoder is learned by enforcing prediction consistency among multiple classifiers, so as to mitigate the overfitting problem and restrict the evolving pattern to be reflected in the classifier as much as possible. During inference, we adopt the ensemble manner based on a great number of target domain-customized classifiers, which are cheaply obtained via the conditional diffusion model, for robust prediction. Comprehensive experiments on both synthetic and real-world datasets show the superior generalization performance of W-Diff on unseen domains in the future.

## 1 Introduction

Domain generalization (DG) deals with a fundamental problem in modern machine learning [13, 46, 47], where performance degeneration often occurs when deep models encounter out-of-distribution (OOD) data [38, 56]. The goal of DG is to learn a model that can perform well on unseen target domains by leveraging labeled data from multiple related but different source domains [23, 19, 57, 22].

---

✉ Corresponding author. Code is available at `https://github.com/BIT-DA/W-Diff`.

38th Conference on Neural Information Processing Systems (NeurIPS 2024).

Despite of abundant works on DG and promising progress so far, they typically embark upon the generalization among stationary and discrete environments [29], where the distribution shift among domains is obvious and remains static over time. In contrast, there have emerged several works on evolving domain generalization (EDG) [24, 29, 51, 2, 44, 50] in recent years, where the data distribution gradually shifts in an underlying continuous structure, e.g., the age-related structural changes in the optic nerve in ocular diseases [12]. But most of EDG methods still assume multiple source domains simultaneously ready, which may be impractical in real world. As the data distribution constantly evolves along time, training data from new data distributions will continue to emerge. Hence, equipping models with lifelong learning ability is crucial for their practical applications.

Despite of the fact that existing researches on continual learning [16, 32, 5] have studied the empowerment of lifelong learning, their focus is on maintaining the performance of seen tasks, instead of generalizing on unseen domains in the future. Therefore, it is still an open problem to achieve evolving domain generalization in the domain-incremental setting, where source domains sequentially arrive to mimic the dynamics of training domains. To this end, previous work EvoS [45] models the features from each domain as a Gaussian distribution and proposes to capture the evolving pattern at the feature level by leveraging self-attention mechanism to generate the feature mean and variance for future domain based on those of historical domains. However, the assumption that features adhere to a Gaussian distribution may not always be applicable. Different from EvoS, we propose to excavate evolving pattern at the parameter level and further achieve domain-customized parameter generation.

Inspired by neural network diffusion [42] that there exist specific parameter patterns in optimized model layers and these patterns can be modeled with diffusion model, we propose to capitalize on the strong modeling ability of diffusion models to capture the evolving pattern of optimized classifiers across domains. To achieve this, we propose a *Weight Diffusion (W-Diff)* approach, which is specialized for EDG in the domain-incremental setting. Specifically, to address the problem of inaccessible historical data, we maintain a first-in-first-out (FIFO) queue to store the optimized classifier weights of historical domains. The stored classifier weights of each historical domain serve as a *reference point* to calculate the change between the classifier weights of current domain *(regarded as the anchor point)* and that of corresponding historical domain. The changes of classifier weights between the anchor point and different reference points provide evolving patterns at different time intervals, which can be utilized to make the modeling of evolving patterns more robust.

In addition, for guidance on how to switch from a reference point to the anchor point, we condition the diffusion model on the class prototypes of current domain along with the reference point. Then, the conditional diffusion model is trained to model the distribution of *residual classifier weights*, i.e., the change of classifier weights between the anchor point and reference point. Meanwhile, to reduce the overfitting of the feature encoder and to restrict the evolutionary pattern to be reflected only in the classifier as much as possible, we learn a domain-shared feature space by enforcing the predictions from different classifiers to be consistent. Finally, during the inference stage, we adopt weights ensemble to give robust predictions based on a great number of generated classifier weights by the diffusion model that is conditioned on current class prototypes and different reference points.

**Contributions: 1)** We study the under-explored area of evolving domain generalization in the domain-incremental setting and explore the innovative usage of diffusion model for this problem. **2)** We propose a novel weight diffusion (W-Diff) approach to capture the evolutionary pattern at the parameter level, orthogonal to previous feature level approaches. Capitalizing on the strong generative ability of diffusion model, W-Diff can generate customized parameters by controlling the condition and make robust predictions via weights ensemble. **3)** Comprehensive experiments on both synthetic and real-world datasets verify the effectiveness and superiority of W-Diff on generalization.

## 2 Related Work

**Evolving Domain Generalization (EDG)** learns the evolving pattern underlying in multiple source domains to achieve generalization capability on the unseen future target domains over time [2, 50, 24, 52, 44, 29]. To name a few, SDE-EDG [50] introduces stochastic differential equations (SDEs) to model the evolving pattern through individual temporal trajectories. DRAIN [2] builds on the Bayesian framework and leverages LSTM to infer the future status of the whole network. However, most of these methods, except for DRAIN, require multiple source domains to be simultaneously available. Very recently, EvoS [45] focuses on EDG with sequentially arriving domains, considering

the low efficiency of training the model from scratch once the accumulated domains are updated. It assumes that features for each domain follow a Gaussian distribution and then models the evolving pattern of the feature distribution, while this assumption is not always suitable. Besides, in the generalization process of multiple consecutive target domains, the statistics generated from previous timestamps are used as inputs to the attention mechanism to generate the statistics at next timestamp, which in turn serve as the input for next generation. This manner is likely to cause error accumulation if previous generation is not accurate. Orthogonal to EvoS [45], we proposes to mine the evolving pattern at the model parameter level and further implement domain-oriented parameter generation via controlling the condition of the diffusion model to avoid the potential error accumulation in EvoS.

**Continual Learning (CL)** focuses on the scenario where the model is trained on a sequence of tasks, and the model is required to adapt to current task and meanwhile maintain the performance on previous tasks [53, 16, 5, 39, 43, 32, 11]. The literature on this field is abundant. Most of the techniques can be categorized into architecture-based [34, 9], representation-based [4, 43, 10], regularization-based [16, 53, 32] and replay-based [5, 39, 11]. In this work, we also continually train models on sequential domains, but the goal is to generalize well on novel domains in the near future.

**Parameter Generation** has been gaining great interests with the rise of diffusion models [42, 8, 21, 54]. For example, p-diff [42] directly generates high-performing neural network parameters from random noises with a standard latent diffusion model, which verifies the feasibility of modeling the parameter distribution via diffusion models. Nevertheless, p-diff uses unconditional diffusion model and can only generate parameters for in-distribution data. G.pt [28] collects the loss, error or return of task model checkpoints during training as the condition for the diffusion model. However, it is designed for a single dataset to which the training data belongs, thus struggling with distribution shifts. D2NWG [36] uses CLIP [30] to extract features for each sample and leverages Set Transformer [18] to generate dataset encoding from these features. Then, the diffusion model is conditioned on the dataset encoding. But labeled training set samples of a new dataset are required to obtain the dataset encoding, which is infeasible in unlabeled target domains. In addition, ProtoDiff [8] and MetaDiff [54] generate prototype classifiers for the meta-test stage by conditioning the diffusion model on the information (e.g., the prototypes in [8] and the gradients in [54]) from the support set. Yet, they concentrate on few-shot learning and the support set requires labeled data, which is unavailable in the target domain of EDG. By contrast, we aims at capturing the evolving pattern among source domains and leveraging it to enable generalization on future target domains without any labeled target data.

## 3 Preliminaries

### 3.1 Problem Formulation

We consider the evolving domain generalization (EDG) in the domain-incremental setting. Formally, during training phase, we are given $T$ sequentially arriving source (training) domains: $\mathcal{S} = \{\mathcal{D}^1, \mathcal{D}^2, \ldots, \mathcal{D}^T\}$, which are collected at timestamps $t_1 < t_2 < \ldots < t_T$, respectively. Each domain is defined as $\mathcal{D}^t = \{\boldsymbol{x}_i^t, y_i^t\}_{i=1}^{N^t}$, $t = 1, \ldots, T$, where $\boldsymbol{x}_i^t$ is the $i$-th sample from the $t$-th domain, $y_i^t \in \{0, 1, \ldots, C-1\}$ is the category label of sample $\boldsymbol{x}_i^t$, and $N^t, C$ are the number of training samples in the $t$-th domain and the number of categories, respectively. In the domain-incremental setting, we can only access current domain $\mathcal{D}^t$ at timestamp $t_t$ and historical domains $\{\mathcal{D}^1, \ldots, \mathcal{D}^{t-1}\}$ are unavailable. This takes into account the data storage burden, privacy concerns and the dynamic evolution of the source domain. Following previous EDG works [24, 29, 2], the label set is the same among domains, but the data distribution of domains is assumed to continuously evolve over time in some patterns. And our goal is to generalize the model, which is composed of a feature encoder $E_{\boldsymbol{\psi}}$ parameterized with $\boldsymbol{\psi}$ and a classifier $H_{\mathbf{W}}$ parameterized with $\mathbf{W}$, on unseen $K$ target (testing) domains in the future: $\mathcal{T} = \{\mathcal{D}^{T+1}, \ldots, \mathcal{D}^{T+K}\}$. To tackle this problem, we propose to model the evolving pattern at the parameter level via the conditional diffusion model and generate customized parameters for future domains by controlling the condition of the diffusion model.

### 3.2 Diffusion Model

Diffusion models have achieved tremendous success in computer vision by modeling the probability transformation from a prior Gaussian distribution to the target distribution [14, 40]. They typically

comprise a diffusion process to progressively add Gaussian noise to data in a multi-step Markov chain and a denoising process to recover data from the noise via reversing the diffusion process.

**Diffusion process.** Given a clean data point $\boldsymbol{x}_0$ sampled from a real data distribution $q(\boldsymbol{x})$, i.e., $\boldsymbol{x}_0 \sim q(\boldsymbol{x})$, the diffusion process is characterized as a Markov chain which slowly adds random Gaussian noise to $\boldsymbol{x}_0$ in $S$ steps, obtaining a sequence of noisy samples: $\boldsymbol{x}_1, \ldots, \boldsymbol{x}_S$. Formally, this process is expressed as

$$q(\boldsymbol{x}_{1:S}|\boldsymbol{x}_0) = \prod_{s=1}^{S} q(\boldsymbol{x}_s|\boldsymbol{x}_{s-1}), \quad q(\boldsymbol{x}_s|\boldsymbol{x}_{s-1}) = \mathcal{N}(\boldsymbol{x}_s; \sqrt{1-\beta_s}\boldsymbol{x}_{s-1}, \beta_s \mathbf{I}), \tag{1}$$

where $\{\beta_s \in (0,1)\}_{s=1}^{S}$ is a variance schedule, $\mathcal{N}$ represents Gaussian distribution, and $\mathbf{I}$ is the identity matrix. And the forward diffused sample at step $s$, denoted as $\boldsymbol{x}_s$, can be directly obtained in a single step by Eq. (2) without iteratively adding noise:

$$\boldsymbol{x}_s = \sqrt{\bar{\alpha}_s}\boldsymbol{x}_0 + \sqrt{1-\bar{\alpha}_s}\boldsymbol{\epsilon}, \quad \boldsymbol{\epsilon} \sim \mathcal{N}(\mathbf{0}, \mathbf{I}), \tag{2}$$

where $\bar{\alpha}_s = \prod_{s'=1}^{s}(1 - \beta_{s'})$. When step size $S$ approaches infinity, $\boldsymbol{x}_S$ is equivalent to a data point from an isotropic Gaussian distribution, i.e., the prior Gaussian distribution $\mathcal{N}(\mathbf{0}, \mathbf{I})$.

**Denoising process.** Given a start noise $\boldsymbol{x}_S \sim \mathcal{N}(\mathbf{0}, \mathbf{I})$, the denoising process moves backward on the multi-step Markov chain as $s$ decreases from $S$ to 1 to remove the noise at each step $s$, finally recovering the clean data. Concretely, the formulation of the denoising process at step $s$ is denoted as

$$\boldsymbol{x}_{s-1} = \boldsymbol{\mu_\theta}(\boldsymbol{x}_s, s) + \sigma_s \boldsymbol{\epsilon} = \frac{1}{\sqrt{1-\beta_s}}\left(\boldsymbol{x}_s - \frac{\beta_s}{\sqrt{1-\bar{\alpha}_s}}\mathcal{E}_{\boldsymbol{\theta}}(\boldsymbol{x}_s, s)\right) + \sigma_s \boldsymbol{\epsilon}, \tag{3}$$

where $\boldsymbol{\mu_\theta}(\boldsymbol{x}_s, s) = \frac{1}{\sqrt{1-\beta_s}}\left(\boldsymbol{x}_s - \frac{\beta_s}{\sqrt{1-\bar{\alpha}_s}}\mathcal{E}_{\boldsymbol{\theta}}(\boldsymbol{x}_s, s)\right)$ and $\mathcal{E}_{\boldsymbol{\theta}}(\cdot, \cdot)$ is a denoising model parameterized with $\boldsymbol{\theta}$ to estimate the noise. $\sigma_s$ is a variance hyperparameter that is theoretically set to $\sigma_s^2 = \beta_s$ in most diffusion works [14, 26]. During the training stage, the denoising model $\mathcal{E}_{\boldsymbol{\theta}}$ is trained by minimizing the following loss $\mathcal{L}_{diff}$ to minimize the noise estimation error:

$$\mathcal{L}_{diff} = \mathbb{E}_{\boldsymbol{x}_0, s, \boldsymbol{\epsilon}}\left[\|\boldsymbol{\epsilon} - \mathcal{E}_{\boldsymbol{\theta}}(\sqrt{\bar{\alpha}_s}\boldsymbol{x}_0 + \sqrt{1-\bar{\alpha}_s}\boldsymbol{\epsilon}, s)\|^2\right]. \tag{4}$$

**Conditional diffusion model.** The way of conditional diffusion models to generate samples is analogous to the unconditional one, except for the added conditional information. Specifically, as in most conditional diffusion works [25, 17], the denoising model $\mathcal{E}_{\boldsymbol{\theta}}(\boldsymbol{x}_s, s)$ is replaced with $\mathcal{E}_{\boldsymbol{\theta}}(\boldsymbol{x}_s, s, \mathfrak{c})$, where $\mathfrak{c}$ denotes the condition, e.g., class labels, texts, images, etc. The matched condition $\mathfrak{c}$ regulates the sample generation in a supervised manner to ensure the desired image content. And inspired by the generating of specific image contents via introducing conditional information to diffusion models, we propose to achieve domain-oriented parameter generation by controlling the diffusion condition.

## 4   Methodology

With the preliminary knowledge of EDG in the domain-incremental setting and diffusion models, we will present the details of Weight Diffusion (W-Diff) in this section. We begin with how to obtain the data for diffusion model training in Section 4.1 and then model the evolving pattern of parameters via the conditional diffusion model in Section 4.2. Finally, the inference procedure of W-Diff to generate customized classifiers is presented in Section 4.3. The overview of W-Diff is illustrated in Fig. 1.

### 4.1   Per-domain Parameter Fitting in Domain-Incremental Setting

In our approach, we try to capture the evolving pattern in optimized model parameters across domains and further generate customized parameters for target domain via leveraging the learned pattern. Considering the unaffordable training cost if modeling the whole parameters for relatively large models, we choose to excavate the evolutionary pattern in the task-specific head, e,g., the classifier for classification tasks. Nevertheless, the remaining parts of the task model would overfit to current domain and cause degraded generalization if without any processing. To avoid this, we learn a domain-shared feature encoder for all domains and a domain-specific classifier for each domain during the domain-incremental training. As $t$ increases from 1 to $T$, once the training stage on

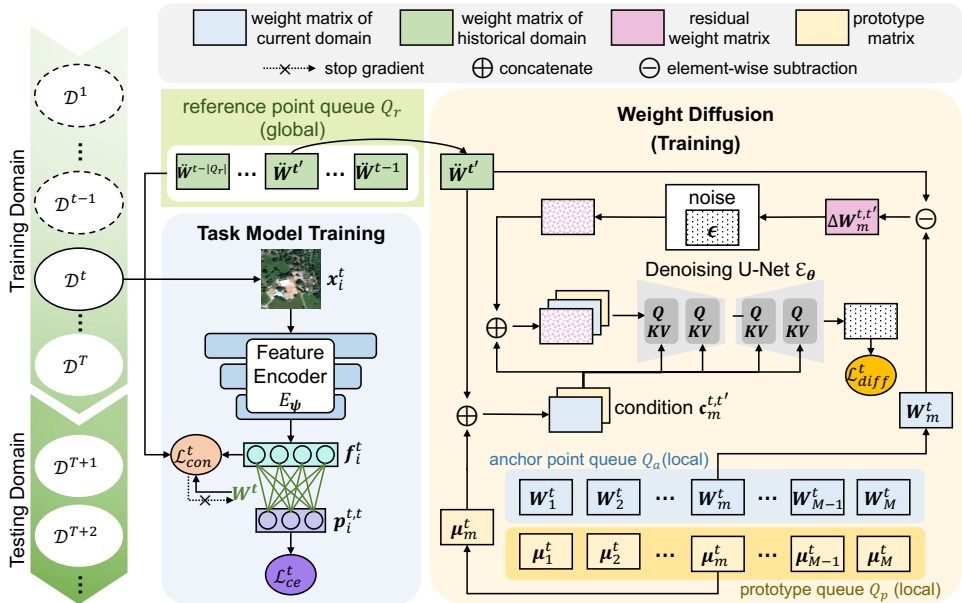

Figure 1: Overview of W-Diff. The reference point queue $Q_r$ stores classifier weights of recent $|Q_r|$ historical domains, and the anchor point queue $Q_a$ and prototype queue $Q_p$ store the updated classifier weights and prototype matrix at each iteration after the warm-up stage on current domain. $\mathcal{L}_{con}^t$ is the prediction consistency loss to learn a domain-shared feature space and $\mathcal{L}_{ce}^t$ is the cross-entropy loss. The conditional diffusion model $\mathcal{E}_{\boldsymbol{\theta}}$ is trained with the noise estimation error loss $\mathcal{L}_{diff}^t$ to model the evolving pattern of classifiers, conditioned on historical reference point and current prototype matrix.

domain $\mathcal{D}^t$ ends, the classifier weights with the best validation performance on the validation set of $\mathcal{D}^t$, denoted as $\ddot{\mathbf{W}}^t \in \mathbb{R}^{C \times d_f}$, is stored in the reference point queue $Q_r$, where $d_f$ the dimension of deep features output by the feature encoder $E_{\boldsymbol{\psi}}$. $Q_r$ is a global FIFO queue with the maximum length $L$ and is used to calculate the change of classifier weights between current domain and a given historical domain in Section 4.2, which reflects the evolving pattern at the parameter level.

**Learning Domain-Shared Feature Encoder.** In the domain-incremental setting, we can only access the data from current domain, which prohibits us from utilizing conventional DG methods that require to access multiple domains simultaneously to learn domain-invariant feature representations. To tackle this problem, we resort to the stored different classifiers in $Q_r$. Intuitively, if domain-shared feature representation is learned, classifiers from different domains should give similar predictions for a given data sample. Hence, at timestamp $t_t$, we train the task model on the $t$-th domain $\mathcal{D}^t$ by minimizing the consistency loss $\mathcal{L}_{con}^t$ to learn a domain-shared feature space:

$$\mathcal{L}_{con}^t = \frac{1}{1+|Q_r|} \cdot \frac{1}{N^t} \cdot \frac{1}{C} \sum_{t'=t-|Q_r|}^{t} \sum_{i=1}^{N^t} KL(\bar{\boldsymbol{p}}_i^t \| \boldsymbol{p}_i^{t,t'}),$$

$$\bar{\boldsymbol{p}}_i^t = \frac{1}{1+|Q_r|} \sum_{t'=t-|Q_r|}^{t} \boldsymbol{p}_i^{t,t'}, \quad \boldsymbol{p}_i^{t,t'} = \begin{cases} \text{softmax}(\text{sg}(\mathbf{W}^t) \times \boldsymbol{f}_i^t), & t' = t \\ \text{softmax}(\text{sg}(\ddot{\mathbf{W}}^{t'}) \times \boldsymbol{f}_i^t), & t' < t \end{cases}, \quad (5)$$

where $\boldsymbol{f}_i^t = E_{\boldsymbol{\psi}}(\boldsymbol{x}_i^t) \in \mathbb{R}^{d_f}$, $|\cdot|$ is the length of the object and $\text{sg}(\cdot)$ denotes stopping gradients. $\ddot{\mathbf{W}}^{t'}$ is the stored classifier weights of historical domain $\mathcal{D}^{t'}$ and $\mathbf{W}^t$ is current classifier weights on $\mathcal{D}^t$.

**Learning Domain-Specific Classifier.** As $t$ increases from 1 to $T$, domain-specific classifier is directly learned by incrementally training the task model via the supervision loss $\mathcal{L}_{ce}^t$ on domain $\mathcal{D}^t$:

$$\mathcal{L}_{ce}^t = \frac{1}{N^t} \sum_{i=1}^{N^t} CrossEntropy \left( \text{softmax}(\mathbf{W}^t \times \boldsymbol{f}_i^t), y_i^t \right). \quad (6)$$

Overall, when training on domain $\mathcal{D}^t$, the task model is optimized by the following total loss $\mathcal{L}_{total}^t$:

$$\mathcal{L}_{total}^t = \mathcal{L}_{ce}^t + \lambda \mathcal{L}_{con}^t, \quad (7)$$

where $\lambda$ is a tradeoff hyperparameter to balance the two losses.

**Collecting Data for Diffusion Model Training.** Considering that parameters at the early stage are unstable, we start to collect the training data for diffusion model after the warm-up stage of the task model is over on corresponding domain. Specifically, when training on the $t$-th domain, the warm-up epochs of the task model account for $\rho \in (0, 1)$ of the total training epochs on domain $\mathcal{D}^t$, where $\rho$ is a hyperparameter. After the warm-up stage, the updated classifier weights $\mathbf{W}^t \in \mathbb{R}^{C \times d_f}$ via back-propagating the gradients of $\mathcal{L}_{total}^t$ and the updated prototype matrix $\boldsymbol{\mu}^t \in \mathbb{R}^{C \times d_f}$ via Eq. (8) are respectively stored into the anchor point queue $Q_a$ and prototype queue $Q_p$ at each iteration.

$$\boldsymbol{\mu}^t[c] := \frac{n^t \cdot \boldsymbol{\mu}^t[c] + \sum_{i=1}^{B} \boldsymbol{p}_i^{t,t}[c] \cdot \boldsymbol{f}_i^t}{n^t + B}, c = 0, 1, \ldots, C-1,$$
$$n^t := n^t + B. \tag{8}$$

$\boldsymbol{\mu}^t[c]$ is the $c$-th row of $\boldsymbol{\mu}^t$, i.e., the prototype of class $c$ on domain $\mathcal{D}^t$ based on all data in the seen batches after the warm-up stage on $\mathcal{D}^t$. $B$ is the batch size of task model. $n^t$ counts the total number of samples in seen batches after the warm-up stage and is initialized to $0$ at the start of training on each domain. $\boldsymbol{p}_i^{t,t}[c]$ is the predicted probability of $\boldsymbol{x}_i^t$ belonging to class $c$ via current classifier of domain $\mathcal{D}^t$. Here, using predicted probability instead of ground-truth label to compute prototypes avoids the issue of missing categories in a batch and the inaccessibility of labels in testing domains.

In implementation, $Q_a$ and $Q_p$ are both FIFO queue with the maximum length $M$, i.e., the training batch size of the diffusion model. When they are full, the training of the conditional diffusion model starts. Note that $Q_a$ and $Q_p$ are only used during the training phase and are locally used on each domain. That is, they are initialized to empty at the beginning of the training stage on each domain.

## 4.2 Modeling Parameter Evolution Pattern with Conditional Diffusion Model

Having prepared the data for diffusion model training, we can utilize them to learn the evolving pattern of parameters during the domain-incremental training process of the task model. To be specific, we use the difference between the classifier weights of current domain and that of a given historical domain to represent the evolution of parameters:

$$\Delta \mathbf{W}_m^{t,t'} = \mathbf{W}_m^t - \ddot{\mathbf{W}}^{t'}, \tag{9}$$

where $\mathbf{W}_m^t$, $m = 1, 2, \ldots, M$, is the classifier weights of current domain $\mathcal{D}^t$, which is cached in the anchor point queue $Q_a$ when training the task model on $\mathcal{D}^t$. And $\ddot{\mathbf{W}}^{t'}$ is the classifier weights of the historical domain $\mathcal{D}^{t'}$ from the reference point queue $Q_r$, $t' \in \{t - |Q_r|, \ldots, t - 1\}$. Hence, the residual classifier weights $\{\Delta \mathbf{W}_m^{t,t'}\}_{m=1}^M$ represent how to evolve from a reference point to the anchor point. Moreover, to guide the evolution, we provide the paired condition for each residual classifier weight matrix $\Delta \mathbf{W}_m^{t,t'}$, where the paired condition is formulated as $\mathfrak{c}_m^{t,t'} = \ddot{\mathbf{W}}^{t'} \oplus \boldsymbol{\mu}_m^t \in \mathbb{R}^{C \times d_f \times 2}$ and $\oplus$ denotes concatenating. The additional condition provides the information about the the starting point and knowledge about the distribution of data in the feature space, which is rightly the anchor point, i.e., the optimized classifier, needs to adapt to. Then, when the task model is incrementally trained on the $t$-th domain, the conditional diffusion model is also incrementally trained to minimize the following noise estimation error loss $\mathcal{L}_{diff}^t$ in Eq. (10), so as to learn how to generate the desired residual classifier weights when given the reference point and prototype matrix as the condition.

$$\mathcal{L}_{diff}^t = \mathbb{E}_{\ddot{\mathbf{W}}^{t'} \in Q_r, \mathbf{W}_m^t \in Q_a, \boldsymbol{\epsilon} \sim \mathcal{N}(\mathbf{0}, \mathbf{I}), s} \left[ \| \boldsymbol{\epsilon} - \mathcal{E}_{\boldsymbol{\theta}}(\sqrt{\bar{\alpha}_s} \cdot \Delta \mathbf{W}_m^{t,t'} + \sqrt{1 - \bar{\alpha}_s} \boldsymbol{\epsilon}, s, \mathfrak{c}_m^{t,t'}) \|^2 \right]. \tag{10}$$

In implementation, the conditional diffusion model adopts the similar U-Net structure as LDM [31] and uses a hybrid conditioning way, i.e., the condition is injected both in the cross-attention and input sides. In Eq. (10), different reference points could enrich the diversity of training data for the conditional diffusion model and provide the evolving pattern at different time intervals.

## 4.3 Generating Customized Classifiers in Inference Phase

After finishing the training on domain $\mathcal{D}^T$, we can use the conditional diffusion model to generate customized classifiers for a given testing domain $\mathcal{D}^{test}$. Firstly, we calculate the prototype matrix $\boldsymbol{\mu}^{test}$ of domain $\mathcal{D}^{test}$ via $\boldsymbol{\mu}^{test}[c] = \frac{1}{N^{test}} \sum_{i=1}^{N^{test}} \bar{\boldsymbol{p}}_i^{test}[c] \cdot \boldsymbol{f}_i^{test}$, where $\bar{\boldsymbol{p}}_i^{test} = \frac{1}{|Q_r|} \sum_{\ddot{\mathbf{W}}^{t'} \in Q_r} \text{softmax}(\ddot{\mathbf{W}}^{t'} \times \boldsymbol{f}_i^{test})$, $c = 0, 1, \ldots, C-1$. Here, we use the more robust average

prediction of multiple classifiers to compute the prototype matrix in the inference phase. Then, given each reference point $\ddot{\mathbf{W}}^{t'}$ in $Q_r$ along with the prototype matrix $\boldsymbol{\mu}^{test}$, we can generate $M_g$ residual classifier weights: $\{\Delta\mathbf{W}_j^{test,t'}\}_{j=1}^{M_g}$ by substituting the denoising net in Eq. (3) with its conditional version and applying the denoising process with condition $\mathfrak{c}^{test,t'} = \ddot{\mathbf{W}}^{t'} \oplus \boldsymbol{\mu}^{test}$. Ultimately, we use the following average weight ensemble $\bar{\mathbf{W}}^{test}$ as the final classifier for label predicting on $\mathcal{D}^{test}$:

$$\bar{\mathbf{W}}^{test} = \frac{1}{|Q_r|}\frac{1}{M_g}\sum\nolimits_{\ddot{\mathbf{W}}^{t'}\in Q_r}\sum\nolimits_{j=1}^{M_g}(\ddot{\mathbf{W}}^{t'} + \Delta\mathbf{W}_j^{test,t'}). \tag{11}$$

In this way, we capitalize on the powerful modeling and generating ability of conditional diffusion model to cheaply produce a great number of target-customized classifiers, which offers more robust and accurate predictions. The pseudo codes of training and testing procedures are in Appendix C.

## 5 Experiments

### 5.1 Experimental Setup

**Benchmark Datasets.** We evaluate W-Diff on both synthetic and real-world datasets [2, 48], including two text classification datasets (**Huffpost**, **Arxiv**), three image classification datasets (**Yearbook**, **RMNIST**, **fMoW**) and two multivariate classification datasets (**2-Moons**, **ONP**). Except for synthetic datasets 2-Moons and RMNIST that use the rotation angle as a proxy for time, all other datasets collect real-world data with the distribution shift over time. Following [45], the number of source and target domains is set as Yearbook: $(T = 16, K = 5)$, RMNIST: $(T = 6, K = 3)$, fMoW: $(T = 13, K = 3)$, Huffpost: $(T = 4, K = 3)$, Arxiv: $(T = 9, K = 7)$, 2-Moons: $(T = 9, K = 1)$, ONP: $(T = 5, K = 1)$. For each source domain, we randomly divide the data into training and validation sets in the ratio of $9 : 1$. For more details on datasets, please refer to Appendix D.1.

**Network Details.** For the task model, we follow the usage in [48, 45]. For the conditional diffusion model, we implement it in a U-Net similar to LDM [31]. Please refer to Appendix D.2 for details.

**Training Details.** For all datasets, we set the batch size $B = 64$, the loss tradeoff $\lambda = 10$ and the maximum length $L = 8$ for the reference point queue $Q_r$. To optimize the task model, we adopt the Adam optimizer with momentum 0.9. As for the warm-up hyperparameter $\rho$, we $\rho = 0.6$ for Huffpost, fMoW and $\rho = 0.2$ for Arxiv, Yearbook, RMNIST, 2-Moons, ONP. For the conditional diffusion model, we set the maximum diffusion step $S = 1000$ and use the AdamW optimizer with batch size $M = 32$, where $M$ is also the maximum length of queue $Q_a$ and $Q_p$. And the number of generated residual classifier weights based on each reference point is set to $M_g = 32$ in the inference stage. All experiments are conducted using the PyTorch packages and run on a single NVIDIA GeForce RTX 4090 GPU with 24GB memory. Three independent experiments with different random seeds are repeated for each task to report the mean and standard deviation (std) of accuracy, which is denoted in the format of "mean ± std" in the table. Please refer to Appendix D.3 for more details.

Table 1: Accuracy (%) on Huffpost and Arxiv. The best and second best results in the incremental setup are bolded and underlined, respectively. (Huffpost: $K = 3$, Axriv: $K = 7$)

| Method | Incremental training | Access multiple domains | Huffpost Accuracy (%) ↑ | | | Arxiv Accuracy (%) ↑ | | |
|---|---|---|---|---|---|---|---|---|
| | | | $\mathcal{D}^{T+1}$ | OOD avg. | OOD worst | $\mathcal{D}^{T+1}$ | OOD avg. | OOD worst |
| Offline | ✗ | ✓ | 72.74 | 71.50 | 69.63 | 57.49 | 52.38 | 49.28 |
| IRM [1] | ✗ | ✓ | 71.04 | 70.31 | 68.97 | 51.11 | 45.89 | 42.86 |
| CORAL [37] | ✗ | ✓ | 71.34 | 70.08 | 68.68 | 50.98 | 45.77 | 42.71 |
| Mixup [55] | ✗ | ✓ | 73.34 | 71.16 | 69.29 | 57.58 | 52.77 | 49.62 |
| LISA [49] | ✗ | ✓ | 72.19 | 70.24 | 68.60 | 56.53 | 52.41 | 49.67 |
| GI [24] | ✗ | ✓ | 68.06 | 66.32 | 64.64 | 53.43 | 49.19 | 46.13 |
| IncFinetune | ✓ | ✗ | 73.57 | 71.98 | 69.80 | 56.22 | 52.43 | 49.37 |
| Mixup [55] | ✓ | ✗ | 73.07 | 71.52 | 69.44 | _56.64_ | 52.95 | 49.97 |
| EWC [16] | ✓ | ✗ | _73.64_ | 71.53 | 68.99 | 56.60 | 52.78 | 49.73 |
| SI [53] | ✓ | ✗ | 72.58 | 71.50 | 69.61 | 49.98 | 47.27 | 44.77 |
| A-GEM [5] | ✓ | ✗ | 72.23 | 71.16 | 69.10 | 52.02 | 48.91 | 46.03 |
| DRAIN [2] | ✓ | ✗ | 73.42 | 71.75 | 69.69 | 56.04 | 52.07 | 48.97 |
| EvoS [45] | ✓ | ✗ | 73.42 | **72.36** | _70.19_ | 56.60 | _53.15_ | _50.19_ |
| **W-Diff** | ✓ | ✗ | **73.91**±0.19 | _72.29_±0.14 | **70.40**±0.12 | **56.66**±0.11 | **53.43**±0.10 | **50.70**±0.20 |

Table 2: Accuracy (%) on Yearbook, RMNIST and fMoW. The best and second best results in the incremental setup are bolded and underlined. (Yearbook: $K = 5$, RMNIST: $K = 3$, fMoW: $K = 3$)

| Method | Incremental training | Access multiple domains | Yearbook Accuracy (%) ↑ | | | RMNIST Accuracy (%) ↑ | | | fMoW Accuracy (%) ↑ | | |
|---|---|---|---|---|---|---|---|---|---|---|---|
| | | | $\mathcal{D}^{T+1}$ | OOD avg. | OOD worst | $\mathcal{D}^{T+1}$ | OOD avg. | OOD worst | $\mathcal{D}^{T+1}$ | OOD avg. | OOD worst |
| Offline | ✗ | ✓ | 89.30 | 88.46 | 86.81 | 98.15 | 92.14 | 83.89 | 72.43 | 59.76 | 49.85 |
| IRM [1] | ✗ | ✓ | 97.09 | 94.52 | 92.58 | 95.10 | 85.05 | 72.52 | 64.77 | 54.92 | 46.51 |
| CORAL [37] | ✗ | ✓ | 95.94 | 91.79 | 88.84 | 93.04 | 79.10 | 62.96 | 62.14 | 51.42 | 42.19 |
| Mixup [55] | ✗ | ✓ | 94.98 | 91.12 | 88.35 | 97.11 | 89.66 | 79.63 | 70.27 | 57.73 | 48.04 |
| LISA [49] | ✗ | ✓ | 95.51 | 92.97 | 91.29 | 96.21 | 87.04 | 75.15 | 70.05 | 55.52 | 44.61 |
| CDOT [27] | ✗ | ✓ | 95.17 | 92.90 | 91.46 | 97.96 | 90.19 | 79.67 | - | - | - |
| CIDA [41] | ✗ | ✓ | 92.36 | 90.67 | 88.45 | 97.43 | 89.19 | 78.32 | - | - | - |
| GI [24] | ✗ | ✓ | 97.42 | 96.37 | 95.73 | 97.78 | 91.00 | 82.46 | 61.62 | 50.83 | 42.78 |
| LSSAE [29] | ✗ | ✓ | 93.93 | 92.12 | 88.75 | 96.73 | 90.36 | 82.13 | 59.15 | 48.66 | 41.38 |
| IncFinetune | ✓ | ✗ | 96.61 | 94.72 | 93.48 | 98.62 | 92.80 | 84.61 | 65.52 | 53.99 | 45.23 |
| Mixup [55] | ✓ | ✗ | 90.21 | 89.83 | 88.43 | 98.43 | 92.38 | 83.45 | 64.84 | 52.00 | 42.54 |
| SimCLR [6] | ✓ | ✗ | 95.94 | 93.07 | 89.65 | 98.23 | 90.98 | 81.05 | 64.97 | 53.20 | 44.71 |
| SwAV [3] | ✓ | ✗ | **97.37** | 94.27 | 91.44 | 98.08 | 90.85 | 80.96 | 66.47 | 54.51 | 45.29 |
| EWC [16] | ✓ | ✗ | 97.18 | 95.12 | 93.64 | 98.56 | 92.02 | 82.80 | 66.23 | 54.55 | 45.80 |
| SI [53] | ✓ | ✗ | 97.09 | 94.67 | 93.48 | 98.61 | 93.27 | 85.65 | 66.61 | 54.89 | 46.46 |
| A-GEM [5] | ✓ | ✗ | 94.36 | 90.96 | 88.88 | 95.99 | 86.95 | 75.45 | 54.54 | 47.61 | 41.13 |
| SGP [32] | ✓ | ✗ | 95.65 | 92.92 | 91.39 | 97.12 | 88.97 | 78.05 | - | - | - |
| DRAIN [2] | ✓ | ✗ | 96.23 | 94.71 | 93.73 | 98.52 | 93.09 | 85.75 | 67.22 | 55.05 | 46.24 |
| EvoS [45] | ✓ | ✗ | **97.37** | **95.53** | **94.78** | 98.64 | 93.84 | 87.04 | 67.18 | 54.64 | 45.86 |
| **W-Diff** | ✓ | ✗ | 97.32±0.23 | 95.03±0.17 | 94.05±0.31 | **98.70±0.04** | **94.12±0.12** | **87.36±0.20** | **68.80±0.19** | **55.86±0.16** | **46.51±0.23** |

Table 3: (a): Error rate (%) on 2-Moons and ONP ($K = 1$). (b): Ablation study on RMNIST.

(a)

| Method | Error rate (%) ↓ | |
|---|---|---|
| | 2-Moons | ONP |
| Offline | 22.4±4.6 | 33.8±0.6 |
| LastDomain | 14.9±0.9 | 36.0±0.2 |
| IncFinetune | 16.7±3.4 | 34.0±0.3 |
| CDOT [27] | 9.3±1.0 | 34.1±0.0 |
| CIDA [41] | 10.8±1.6 | 34.7±0.6 |
| GI [24] | 3.5±1.4 | 36.4±0.8 |
| DRAIN [2] | 3.2±1.2 | 38.3±1.2 |
| EvoS [45] | 2.5±1.0 | 33.1±0.6 |
| **W-Diff** | **1.5±1.0** | **32.9±0.5** |

(b)

| Method | Conditioning way for $\mathcal{E}_{\boldsymbol{\theta}}$ | Loss $\mathcal{L}^t_{con}$ | Computed $\bar{\mathbf{W}}^{test}$ based on * | | | Accuracy (%) ↑ | | |
|---|---|---|---|---|---|---|---|---|
| | | | $Q_r$ | noise-added $Q_r$ | diffusion | $\mathcal{D}^{T+1}$ | OOD avg. | OOD worst |
| variant A | hybrid | - | - | - | ✓ | 98.48 | 91.69 | 81.87 |
| variant B | - | ✓ | - | - | - | 98.55 | 93.70 | 86.91 |
| variant C | - | ✓ | ✓ | - | - | 98.47 | 93.82 | 87.28 |
| variant D | - | ✓ | - | ✓ | - | 98.55 | 93.78 | 87.13 |
| variant E | cross_attn | ✓ | - | - | ✓ | 97.98 | 91.68 | 83.19 |
| variant F | concat | ✓ | - | - | ✓ | 98.44 | 92.93 | 85.03 |
| **W-Diff** | hybrid | ✓ | - | - | ✓ | **98.70** | **94.12** | **87.36** |

**Evaluation Metrics.** We report the generalization performance on $K$ target domains in the future, including the average accuracy "OOD avg." ($\frac{1}{K}\sum_{k=1}^{K} \text{Accuracy}(\mathcal{D}^{T+k})$) and the worst accuracy "OOD worst" ($\min_{k \in \{1,...,K\}} \text{Accuracy}(\mathcal{D}^{T+k})$) on $K$ target domains and the accuracy on $\mathcal{D}^{T+1}$.

## 5.2 Main Results

We provide the quantitative results in Table 1, 2, 3(a), where the results of baselines in the non-incremental and incremental scenarios are reported from [45]. For ONP dataset, we notice that previous continuous domain adaptation methods (CDOT and CIDA) and EDG methods (GI and DRAIN) all perform worse than the Offline method that trains the task model on the cumulation of all source domains. In contrast, our W-Diff still obtains superior accuracy to previous state-of-the-art method (EvoS) on this challenging dataset, which validates the superiority of W-Diff. Besides, our W-Diff also achieves the best results on Huffpost, Arxiv, RMNIST and fMoW, in terms of the OOD worst accuracy. These results benefit from the modeling of parameter evolution pattern and the more robust predictions via the weight ensemble based on the conditional diffusion model. For Yearbook dataset, DRAIN, which models the evolution of whole model parameters via LSTM, is inferior to EvoS, which models the evolution of domain-level feature distribution. It suggests that modeling the evolving pattern at the feature level may be more appropriate for Yearbook, which also explains why W-Diff does not obtain the state-of-the-art performance on Yearbook. But our W-Diff still obviously outperforms DRAIN. Overall, we can observe that W-Diff surpasses the baselines in the incremental-training setup on six out of seven datasets, which shows the superiority of W-Diff.

## 5.3 Analytical Experiments

**Ablation Study.** Firstly, the significant performance drop of variant A in Table 3(b) suggests that learning domain-invariant feature representations is necessary for EDG in the domain-incremental setting. Otherwise, the feature encoder could easily overfit to current domain, prohibiting the task model from generalization. Then, we try different ways to construct the average weight ensemble $\bar{\mathbf{W}}^{test}$, including variant C which directly uses the historical classifier weights in $Q_r$, i.e., $\bar{\mathbf{W}}^{test} = \frac{1}{|Q_r|}\sum_{\ddot{\mathbf{w}}^{t'} \in Q_r} \ddot{\mathbf{W}}^{t'}$, and variant D which augments classifier weights by adding small

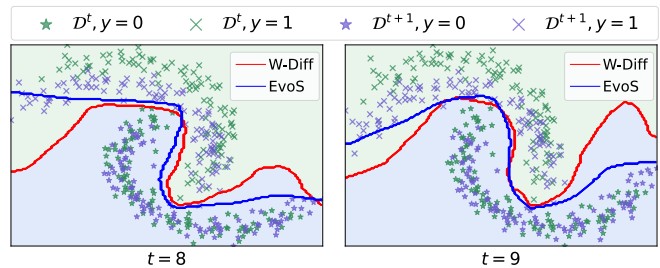
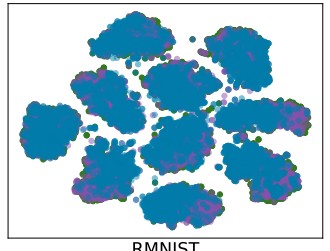

| (a) Visualization of decision boundary on 2-Moons. | (b) t-SNE visualization of features. |

Figure 2: (a): Decision boundary of EvoS [45] and W-Diff on 2-Moons, where we incrementally train the model until the $t$-th domain and then visualize the decision boundary for future domain $\mathcal{D}^{t+1}$. (b): visualization of features from target domains, where different colors represent different domains.

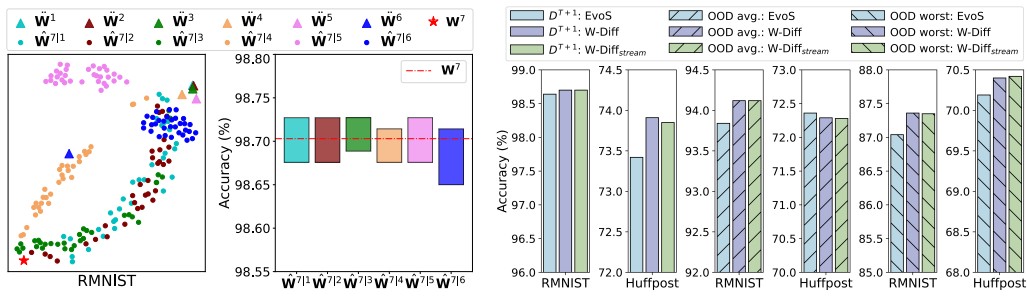

| (a) Visualization of classifier weights. | (b) Results when evaluating in batch-data stream. |

Figure 3: (a): Visualization of classifier weights for $\mathcal{D}^{T+1}, T = 6$, on RMNIST and their accuracy range. $\ddot{\mathbf{W}}^{t'}, t' = 1, \ldots, 6$, is the reference point from $Q_r$, $\hat{\mathbf{W}}^{7|t'}$ is the generated $M_g$ classifier weights based on $\ddot{\mathbf{W}}^{t'}$, and $\mathbf{W}^7$ is the average weights of $\mathcal{D}^7$ fine-tuned classifier weights in the last 200 iterations. (b): Accuracy of EvoS and W-Diff on RMNIST and Huffpost datstes, where W-Diff$_{stream}$ denotes W-Diff is evaluated with batch-data stream for each target domain.

noises to the weights in $Q_r$, i.e., $\bar{\mathbf{W}}^{test} = \frac{1}{|Q_r|} \frac{1}{M_g} \sum_{\ddot{\mathbf{W}}^{t'} \in Q_r} \sum_{j=1}^{M_g} (\ddot{\mathbf{W}}^{t'} + noise_j), noise_j \sim$ Uniform$(-0.01, 0.01)$. The inferior results of variant B, C and D indicate that W-Diff benefits from generating meaningful and customized classifier weights via controlling the condition of diffusion model. Finally, different conditioning ways for the diffusion model are explored, including variant E which injects the condition only in the cross-attention, and variant F which injects the condition only in the input by concatenating the condition with diffused residual classifier weights. Results of variant E and F are both unsatisfactory. This is probably due to the large gap between the residual classifier weights in the input side and the full classifier weights in the condition side, which makes the information interaction hard. And injecting the condition only on the input side can lead to insufficient information interaction. Empirically, we find that the hybrid manner works best.

**Decision Boundary Visualization on Future Domain.** In Fig. 2(a), the model is incrementally trained until the training stage on the $t$-th domain $\mathcal{D}^t$ finishes. Then we visualize the decision boundary on the next future domain $\mathcal{D}^{t+1}, t = 8, 9$. From the results, we can see that the decision boundary of W-Diff adapts to the evolution of domains better than that of EvoS, which shows the superiority of W-Diff in addressing evolving domain generalization in the domain-incremental setup.

**t-SNE Visualization of Features.** In this qualitative experiment, we visualize the features of future target domains for RMNIST dataset to show the effectiveness of $\mathcal{L}_{con}^t$. From Fig. 2(b), we can observe that features from different target domains align well. To some extent, it verifies the effectiveness of $\mathcal{L}_{con}^t$ to learn a domain-shared feature space, which contributes to the mitigation of distribution shift.

**Visualization of Generated Classifier Weights.** In Fig. 3(a), we plot the generated classifier weights for domain $\mathcal{D}^{T+1}, T = 6$, on RMNIST, as well as the accuracy range on $\mathcal{D}^{T+1}$ of generated classifier weights based on different reference points. In Fig. 3(a), some generated weights locate close to the average fine-tuned weights $\mathbf{W}^7$, showing that W-Diff generates domain-customized classifiers.

Table 4: Accuracy (%) of W-Diff on RMNIST dataset using different conditions. ($K = 3$)

| Method | Condition | Accuracy (%) ↑ | | |
|--------|-----------|------|------|------|
| | | $\mathcal{D}^{T+1}$ | OOD avg. | OOD worst |
| W-Diff | reference point $\oplus$ prototype matrix | **98.70** | 94.12 | 87.36 |
| W-Diff | scaled reference point$^\ddagger$ $\oplus$ prototype matrix | 98.69 | **94.17** | **87.46** |

$^\ddagger$ denotes the reference point is scaled by the factor $\eta = 1.5 - \frac{1}{1+e^{-\Delta t}}$, where $\Delta t$ is the timestamp difference between the reference point and anchor point.

Besides, $\hat{\mathbf{W}}^{7|1}$, $\hat{\mathbf{W}}^{7|2}$, $\hat{\mathbf{W}}^{7|3}$ are similar, while $\hat{\mathbf{W}}^{7|4}$, $\hat{\mathbf{W}}^{7|5}$, $\hat{\mathbf{W}}^{7|6}$ are different, due to the more pronounced differences among reference points $\ddot{\mathbf{W}}^4$, $\ddot{\mathbf{W}}^5$, and $\ddot{\mathbf{W}}^6$. This implies that the evolution pattern may be different at different time intervals. And these diverse and high-performing generated weights based on different reference points could conduce to more robust predictions.

**Evaluating in Batch-Data Stream.** In addition to the inference way in Section 4.3, we also provide another version, where the data in target domain arrives batch by batch. Concretely, we use the iterative manner in Eq. (8) along with the average prediction in Section 4.3 to update the prototype matrix $\boldsymbol{\mu}^{test}$, once a batch of data from $\mathcal{D}^{test}$ arrives. Then, we compute the average weight ensemble $\bar{\mathbf{W}}^{test}$ via Eq. 11 for this data batch. Fig. 3(b) shows the results on RMNIST and Huffpost. The two manners present similar results and users can choose appropriate manner based on their scenarios.

**Equipping Condition with Timestamp Difference.** In this part, we try to explicitly incorporate the timestamp difference between the anchor point and reference point into the condition of diffusion model. Concretely, we scale the reference point $\ddot{\mathbf{W}}^{t'}$ in the condition $\mathfrak{c}^{t,t'}$ by a factor $\eta$, where $\eta = 1.5 - \frac{1}{1+e^{-\Delta t}}$ and $\Delta t = t_t - t_{t'}$ is the timestamp difference between reference and anchor points. More distant reference points have larger $\Delta t$ and are weakened. The results on RMNIST dataset are given in Table 4, where the average and worst accuracies of target domains improve slightly. The insignificant performance improvement may be due to the fact that our approach implicitly takes the timestamp difference into account via the domain-incremental training and residual classifier weights.

**Results with Larger Backbones.** We try larger backbones on the fMoW dataset by replacing the DenseNet-121 with DenseNet-169/201/161 [15], respectively. The results are provided in Table 5. When applying lagers backbones, our method still works well and further performance improvements are obtained. This benefits from the consideration of only modeling the evolution of classifier weights, instead of the whole network

Table 5: Accuracy (%) of W-Diff on fMoW dataset with different backbones. ($K = 3$)

| Backbones | parameters | Accuracy (%) ↑ | | |
|-----------|-----------|------|------|------|
| | | $\mathcal{D}^{T+1}$ | OOD avg. | OOD worst |
| DenseNet-121 (growth rate=32) | 64.5MB | 68.80 | 55.86 | 46.51 |
| DenseNet-169 (growth rate=32) | 114.4MB | 70.20 | 56.81 | **47.50** |
| DenseNet-201 (growth rate=32) | 161.8MB | 70.38 | 56.28 | 46.40 |
| DenseNet-161 (growth rate=48) | 230.8MB | **71.28** | **57.36** | 47.33 |

parameters. Otherwise, the training difficulty and huge memory burden from the conditional diffusion model would be unbearable, despite of the greater feature extraction capability of larger backbones.

# 6 Conclusion

This work delves into the under-explored problem of evolving domain generalization in the domain-incremental setting, where the source domain is also non-stationary and dynamically evolves. To tackle this, we propose a Weight Diffusion (W-Diff) approach to capture the evolving pattern across domains at the parameter level and further generate customized classifiers for future domains. W-Diff innovatively leverages the conditional diffusion model to learn the evolution of classifiers from historical domain to current domain, conditioned on the historical classifier weights and current prototype matrix. Extensive results on synthetic and real-world datasets verify the efficacy of W-Diff.

# Acknowledgements

This paper was supported by National Key R&D Program of China (No. 2021YFB3301503), the National Natural Science Foundation of China (No. 62376026), and also sponsored by Beijing Nova Program (No. 20230484296), CCF-Tencent Rhino-Bird Open Research Fund and KuaiShou.

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

## Appendix Contents

## A   Broader Impacts & Limitations

**Broader Impacts.** In this work, we explore the evolving domain generalization in the domain incremental setting. The ability to continually learn from dynamic source domains and leverage the learned evolving pattern to generalize on unseen domains in the future may benefit relevant non-stationary scenarios, e.g., advertisement recommendation with continually emerging new training data and autonomous driving with distribution shift over time or geographical position, etc. It reduces the time and cost for labeling data of target domain and avoids the low efficiency of training the model from scratch with all saved domains once new training domain is available. Yet, for high-security demanding scenarios, the prediction from the model should be adopted with caution to avoid severe accidents, as failures can occur in our method when facing significant distribution shifts.

**Limitations.** Our work presents a way to capture the evolving pattern at the parameter level via capitalizing on the powerful modeling ability of conditional diffusion model. Yet, like any research, our work is not absolutely perfect. There are indeed some limitations that should be acknowledged. Firstly, the task considered in this paper limits to the classification. In the future, we may extend our method to more diverse tasks, e.g., regression tasks. Besides, considering the training cost, we only model the evolution of classifiers. Perhaps, it is feasible to consider more parameters by mapping them into a low-dimensional latent space, but achieving the accurate encoding and decoding is not easy. We leave this for a future work.

## B   Notation Table

Given the large number of notations used throughout the paper, we provide an overall notation description in Table 6 to ease the burden on readers.

Table 6: Notation description

| | |
|---|---|
| ***Data-related*** | |
| $T$ | the number of source domains |
| $K$ | the number of target domains |
| $C$ | the number of categories |
| $c$ | the index of categories ($c \in \{0, 1, \ldots, C-1\}$) |
| $t, t'$ | the index of domains ($t, t' \in \{1, 2, \ldots, T+K\}$) |
| $\mathcal{D}^t$ | the $t$-th domain |
| $t_t$ | the timestamp which the $t$-th domain is collected at |
| $N^t$ | the number of samples in the $t$-th domain |
| $\boldsymbol{x}_i^t$ | the $i$-th sample in the $t$-th domain |
| $y_i^t$ | the category label of the $i$-th sample in the $t$-th domain ($y_i^t \in \{0, 1, \ldots, C-1\}$) |
| $\mathcal{S}$ | a sequence of source (training) domains ($\mathcal{S} = \{\mathcal{D}^1, \mathcal{D}^2, \ldots, \mathcal{D}^T\}$) |
| $\mathcal{T}$ | a sequence of target (testing) domains ($\mathcal{T} = \{\mathcal{D}^{T+1}, \ldots, \mathcal{D}^{T+K}\}$) |
| $\mathcal{D}^{test}$ | a testing domain ($\mathcal{D}^{test} \in \mathcal{T}$) |
| ***Task model-related*** | |
| $B$ | batch size of task model |
| $E_{\boldsymbol{\psi}}$ | feature extractor |
| $\boldsymbol{\psi}$ | parameters of the feature extractor |
| $d_f$ | the dimension of of deep features output by the feature encoder |
| $\boldsymbol{f}_i^t$ | the deep features of the $i$-th sample in the $t$-th domain ($\boldsymbol{f}_i^t = E_{\boldsymbol{\psi}}(\boldsymbol{x}_i^t) \in \mathbb{R}^{d_f}$) |
| $H_{\mathbf{W}}$ | classifier |

| | |
|---|---|
| $\mathbf{W}$ | parameters of the classifier |
| $Q_r$ | the reference point queue |
| $L$ | the maximum length of $Q_r$ |
| $\ddot{\mathbf{W}}^{t'}$ | the saved classifier weights in $Q_r$ for the $t'$-th domain ($\ddot{\mathbf{W}}^{t'} \in \mathbb{R}^{C \times d_f}$) |
| $\mathbf{W}^t$ | the current classifier weights of the $t$-th domain ($\mathbf{W}^t \in \mathbb{R}^{C \times d_f}$) |
| $\boldsymbol{\mu}^t$ | the current prototype matrix of the $t$-th domain ($\boldsymbol{\mu}^t \in \mathbb{R}^{C \times d_f}$) |
| $\boldsymbol{\mu}^t[c]$ | the $c$-th row of $\boldsymbol{\mu}^t$ |
| $n^t$ | the total number of samples in seen batches after the warm-up stage on domain $\mathcal{D}^t$ |
| $\boldsymbol{p}_i^{t,t'}$ | the prediction for the $i$-th sample in the $t$-th domain by the classifier of domain $\mathcal{D}^{t'}$ |
| $\boldsymbol{p}_i^{t,t'}[c]$ | the $c$-th element of $\boldsymbol{p}_i^{t,t'}$ |
| $\bar{\boldsymbol{p}}_i^t$ | the average prediction for the $i$-th sample in the $t$-th domain |
| $Q_a$ | the anchor point queue, which stores the classifier weights of current domain |
| $Q_p$ | the prototype queue, which stores the prototype matrices of current domain |
| $M$ | the maximum length of $Q_a$ and $Q_p$ |
| $m$ | the index of the object in $Q_a$ and $Q_p$ ($m \in \{1, 2, \cdots, M\}$) |
| $\boldsymbol{\mu}_m^t$ | the $m$-th prototype matrix in the prototype queue $Q_p$ of the $t$-th domain |
| $\mathbf{W}_m^t$ | the $m$-th classifier weights in the anchor point queue $Q_a$ of the $t$-th domain |
| $\Delta\mathbf{W}_m^{t,t'}$ | residual classifier weights ($\Delta\mathbf{W}_m^{t,t'} = \mathbf{W}_m^t - \ddot{\mathbf{W}}^{t'}$) |
| $\mathfrak{c}_m^{t,t'}$ | condition of the conditional diffusion model ($\mathfrak{c}_m^{t,t'} = \ddot{\mathbf{W}}^{t'} \oplus \boldsymbol{\mu}_m^t \in \mathbb{R}^{C \times d_f \times 2}$) |
| $\rho$ | warm-up hyperparameter ($\rho \in (0, 1)$) |
| $\lambda$ | tradeoff hyperparameter |
| $\mathcal{L}_{con}^t$ | the consistency loss on the $t$-th domain |
| $\mathcal{L}_{ce}^t$ | the cross-entropy loss on the $t$-th domain |
| $\mathcal{L}_{total}^t$ | the total loss on the $t$-th domain ($\mathcal{L}_{total}^t = \mathcal{L}_{ce}^t + \lambda\mathcal{L}_{con}^t$) |
| **Diffusion model-related** | |
| $S$ | diffusion steps |
| $s, s'$ | the index of diffusion steps ($s, s' \in \{1, 2, \ldots, S\}$) |
| $q(\boldsymbol{x})$ | distribution of variable $\boldsymbol{x}$ |
| $\boldsymbol{x}_0$ | the original data point from $q(\boldsymbol{x})$ |
| $\boldsymbol{x}_s$ | the noisy data point at the $s$-th diffusion step |
| $\beta_s$ | the variance used at the $s$-th diffusion step ($\beta_s \in (0, 1)$) |
| $\bar{\alpha}_s$ | the product of all $(1 - \beta_{s'})$ until the $s$-th diffusion step ($\bar{\alpha}_s = \prod_{s'=1}^s (1 - \beta_{s'})$) |
| $\boldsymbol{\epsilon}$ | random noise |
| $\mathcal{E}_{\boldsymbol{\theta}}$ | denoising model |
| $\boldsymbol{\theta}$ | parameters of the denoising model |
| $\sigma_s$ | a variance hyperparameter |
| $\mathfrak{c}$ | condition of the denoising model |
| $\mathcal{L}_{diff}^t$ | noise estimation error loss on the $t$-th domain |
| $M_g$ | the number of generated classifier weights based on each reference point |
| $\boldsymbol{\mu}^{test}$ | the estimated prototype matrix of the testing domain $\mathcal{D}^{test}$ during inference stage |
| $\mathfrak{c}^{test,t'}$ | the condition of diffusion model for domain $\mathcal{D}^{test}$ ($\mathfrak{c}^{test,t'} = \ddot{\mathbf{W}}^{t'} \oplus \boldsymbol{\mu}^{test}$) |
| $\Delta\mathbf{W}_j^{test,t'}$ | the $j$-th generated residual classifier weights, conditioned on $\mathfrak{c}^{test,t'}$ |
| $\bar{\mathbf{W}}^{test}$ | the average ensemble weights for the testing domain $\mathcal{D}^{test}$ |
| **Others** | |
| $\mathbf{I}$ | identity matrix |
| $\mathcal{N}(\cdot, \cdot)$ | Gaussian distribution |
| $\mathbb{E}$ | mathematic expectation |
| $\mathbb{R}$ | real number |
| $\|\cdot\|$ | L2-Norm |
| $|\cdot|$ | the length of an object |
| $KL(\cdot\|\cdot)$ | Kullback-Leibler divergence |
| $\text{sg}(\cdot)$ | stopping gradients of an object |
| $\text{softmax}(\cdot)$ | normalized exponential function |
| $\oplus$ | concatenating operation |
| $i, j, k$ | indices |

## C  Algorithm of W-Diff

The training and testing procedures of W-Diff are presented in Algorithm 1 and 2.

---

**Algorithm 1:** Training procedure for W-Diff

---

**Input:** sequentially arriving source domains $\mathcal{S} = \{\mathcal{D}^1, \mathcal{D}^2, \ldots, \mathcal{D}^T\}$, feature encoder $E_{\boldsymbol{\psi}}$, classifier $H_{\mathbf{W}}$, conditional diffusion model $\mathcal{E}_{\boldsymbol{\theta}}$, reference point queue $Q_r$ with length $L$, anchor point queue $Q_a$ with length $M$, prototype queue $Q_p$ with length $M$, batch size $B$, loss tradeoff hyperparameter $\lambda$, warm-up hyperparameter $\rho$, maximum diffusion step $S$, training iterations $I_{TS}$ of task model, inner iterations $I_{DM}$ for updating diffusion model.

1  Initialize model parameters $\boldsymbol{\psi}$ as $\boldsymbol{\psi}^0$, $\mathbf{W}$ as $\mathbf{W}^0$, $\boldsymbol{\theta}$ as $\boldsymbol{\theta}^0$ and set $Q_r = \emptyset$.
2  **for** $t = 1$ *to* $T$ **do**
3      Set $\boldsymbol{\psi}^t = \boldsymbol{\psi}^{t-1}$, $\mathbf{W}^t = \mathbf{W}^{t-1}$, $\boldsymbol{\theta}^t = \boldsymbol{\theta}^{t-1}$, $Q_a = \emptyset$, $Q_p = \emptyset$, $\boldsymbol{\mu}^t = \mathbf{0}$, $n^t = 0$.
4      **for** $iter = 1$ *to* $I_{TS}$ **do**
5         $\mathcal{L}^t_{total} = 0$.
6         Randomly sample a batch of data $\{\boldsymbol{x}^t_i, y^t_i\}^B_{i=1}$ from domain $\mathcal{D}^t$.
7         Get the deep features of samples: $\{\boldsymbol{f}^t_i = E_{\boldsymbol{\psi}}(\boldsymbol{x}^t_i)\}^B_{i=1}$.
8         Calculate the supervision loss $\mathcal{L}^t_{ce}$ in Eq. 6.
9         $\mathcal{L}^t_{total} + = \mathcal{L}^t_{ce}$.
10        **if** $t > 1$ **then**
11           Calculate the consistency loss $\mathcal{L}^t_{con}$ in Eq. 5.
12           $\mathcal{L}^t_{total} + = \mathcal{L}^t_{con}$.
13        Update $\boldsymbol{\psi}^t$ and $\mathbf{W}^t$ by backpropagating the gradients of $\mathcal{L}^t_{total}$.
14        **if** $(iter > \rho \cdot I_{TS}) \wedge (t > 1)$ **then**
15           Update prototype matrix $\boldsymbol{\mu}^t$ via Eq. (8).
16           Push $\mathbf{W}^t$ into $Q_a$ and $\boldsymbol{\mu}^t$ into $Q_p$: $Q_a \leftarrow \mathbf{W}^t$, $Q_p \leftarrow \boldsymbol{\mu}^t$.
17        **if** $|Q_a| == M$ **then**
18           **for** $inner\_iter = 1$ *to* $\lceil \frac{I_{DM}}{|Q_r|} \rceil$ **do**
19              Sample diffusion step size $s \sim \text{Uniform}(1, \ldots, S)$ and $\boldsymbol{\epsilon} \sim \mathcal{N}(\mathbf{0}, \mathbf{I})$.
20              Calculate the noise estimation error loss $\mathcal{L}^t_{diff}$ via Eq. (10).
21              Update $\boldsymbol{\theta}^t$ by backpropagating the gradients of $\mathcal{L}^t_{diff}$.
22     Push the classifier weights with the best performance on the validation set of $\mathcal{D}^t$, denoted as $\ddot{\mathbf{W}}^t$, into $Q_r$: $Q_r \leftarrow \ddot{\mathbf{W}}^t$.
23 **return** *Final* $\boldsymbol{\psi}^T, \boldsymbol{\theta}^T, Q_r$.

---

## D  Experimental Setup Details

### D.1  Dataset Description

**Huffpost** (license: CC0: Public Domain) from [48] comprises $63,907$ news headlines from the Huffington Post, with the time span from 2012 to 2018. These news headlines belong to 11 categories: "Black Voices", "Business", "Comedy", "Crime", "Entertainment", "Impact", "Queer Voices", "Science", "Sports", "Tech" and "Travel". This dataset reflects changes in news content and style over time. Follwoing [45], the first 4 years are used for training ($T = 4$) and the last 3 years are used for testing ($K = 3$). For each training domain, we randomly divide the data into training set and validation set in the ratio of $9 : 1$.

**Arxiv** (license: CC0: Public Domain) in [48] is a large-scale dataset, including $2,057,952$ paper titles from 2007 to 2022. It reflects the change over time as research fields evolve. The task is to classify a research paper into one of 172 categories based solely on its title. For this dataset, we use data from the first 9 years as source domains ($T = 9$) and data from the last 7 years as target domains ($K = 7$). For each source domain, the data is randomly divided into training set and validation set in the ratio of $9 : 1$.

**Algorithm 2:** Testing procedure for W-Diff

---

**Input:** sequentially arriving target domains $\mathcal{T} = \{\mathcal{D}^{T+1}, \mathcal{D}^{T+2}, \ldots, \mathcal{D}^{T+K}\}$, feature encoder $E_{\psi^T}$, conditional diffusion model $\mathcal{E}_{\theta^T}$, reference point queue $Q_r$, number of categories $C$, batch size $B$, maximum diffusion step $S$, number of generated residual weights $M_g$ based on each reference point.

**1 for** $k = 1$ **to** $K$ **do**

**2**     Set $\mathcal{D}^{test} = \mathcal{D}^{T+k}$.

**3**     Calculate the prototype matrix $\boldsymbol{\mu}^{test}$ vai $\boldsymbol{\mu}^{test}[c] = \frac{1}{N^{test}} \sum_{i=1}^{N^{test}} \bar{\boldsymbol{p}}_i^{test}[c] \cdot \boldsymbol{f}_i^{test}$, where $\bar{\boldsymbol{p}}_i^{test} = \frac{1}{|Q_r|} \sum_{\ddot{\mathbf{W}}^{t'} \in Q_r} \mathrm{softmax}(\ddot{\mathbf{W}}^{t'} \times \boldsymbol{f}_i^{test})$, $\boldsymbol{f}_i^{test} = E_{\psi^T}(\boldsymbol{x}_i^{test})$, $c = 0, \ldots, C-1$.

**4**     **for** $\ddot{\mathbf{W}}^{t'} \in Q_r$ **do**

**5**         Generate $M_g$ residual classifier weights: $\{\Delta\mathbf{W}_j^{test,t'}\}_{j=1}^{M_g}$ by substituting the denoising net in Eq. (3) with $\mathcal{E}_{\theta^T}$ and applying the denoising process with condition $\mathfrak{c}^{test,t'} = \ddot{\mathbf{W}}^{t'} \oplus \boldsymbol{\mu}^{test}$.

**6**     Obtain the average weight $\bar{\mathbf{W}}^{test} = \frac{1}{|Q_r|} \frac{1}{M_g} \sum_{\ddot{\mathbf{W}}^{t'} \in Q_r} \sum_{j=1}^{M_g} (\ddot{\mathbf{W}}^{t'} + \Delta\mathbf{W}_j^{test,t'})$.

**7**     Get the final label predictions on domain $\mathcal{D}^{test}$: $\{\hat{y}_i^{test} = \mathrm{argmax}_c \, \boldsymbol{p}_i^{test}[c]\}_{i=1}^{N^{test}}$, where $\boldsymbol{p}_i^{test} = \mathrm{softmax}(\bar{\mathbf{W}}^{test} \times \boldsymbol{f}_i^{test})$.

**8 return** *Label Predictions* $\{\{\hat{y}_i^{T+k}\}_{i=1}^{N^{T+k}}\}_{k=1}^{K}$.

---

**Yearbook** (MIT license) dataset comes from [48]. It collects $37,189$ grayscale yearbook photos from 128 American high schools, with the time span from 1930 to 2013. The resolution of photos is $32 \times 32$. Photos from different years reflect changes in fashion trends and social norms over the decades. The task is to classify the genders from a yearbook photo. It is worth mentioning that we only use this dataset to evaluate the generalization performance of different methods in classification tasks. Following [45], we group the data into domains at four-year intervals, resulting in 21 domains. And the first 16 domains are used as source domains ($T = 16$), with the last 5 domains as target domains ($K = 5$). For each source domain, we randomly select 90% of samples as the training split and 10% of samples as the validation split. And for each target domain, we evaluate on its all data.

**RMNIST** (license: CC BY-SA 3.0) is constructed from MNIST dataset [7] which contains grayscale images of digits from 0 to 9. The image resolution is $28 \times 28$. RMNIST first randomly divides all data in MNIST into 9 groups and then creates 9 domains by rotating the 9 groups by $0°, 10°, \ldots, 80°$, respectively. The rotation angle is used to simulate the evolving data distribution over time. Following [45], we use the first 6 domains ($T = 6$) as source domains and the last 3 domains as target domains ($K = 3$). Similarly, we split each source domain into training and validation sets in the ratio of $9 : 1$.

**fMoW** (license: `https://github.com/fMoW/dataset/blob/master/LICENSE`) dataset is from [48], which consists of $141,696$ RGB satellite images from 2002 to 2017. The visual features in these satellite images change over time due to human and environmental activities. The image resolution is $224 \times 224$ and the task is to classify the functional purpose of the buildings or land in a image into one of 62 categories. For this dataset, we consider each year as a separate domain. And the first 13 domains are used for training ($T = 13$), while the last 3 domains are used for testing ($K = 3$). The ratio of training data to validation data for each source domain is $9 : 1$.

**2-Moons** (license: `https://github.com/BaiTheBest/DRAIN/blob/main/LICENSE`) from [2] is constructed from 2-entangled moons dataset, where the lower moon with label 0 and the upper moon with label 1 contain 100 data points, respectively. 2-Moons creates 10 domains by counter-clockwise rotating the 200 data points at an interval of $18°$. Similar to RMNIST, the rotation angle simulates the evolving of data distribution. Following [2], the first 9 ($T = 9$) domains are used as source domain and the last domain is used as target domain ($K = 1$).

**Online News Popularity (ONP)** (license: CC BY 4.0) in [2] summarizes a heterogeneous set of features related to the articles published by Mashable in a two-year period. This dataset is divided into 6 domains by time, and the goal is to predict whether an article is popular in social networks based on its features. Following [2], we use the first 5 domains for training ($T = 5$) and the last domain for testing ($K = 1$).

Table 7: Configuration of the U-Net $\mathcal{E}_{\boldsymbol{\theta}}$ on different datasets with hybrid conditioning way.

| Dataset | Huffpost | Arxiv | Yearbook | RMNIST | fMoW | 2-Moons | ONP |
|---|---|---|---|---|---|---|---|
| Input-shape | $11 \times 128 \times 3$ | $172 \times 128 \times 3$ | $2 \times 32 \times 3$ | $10 \times 128 \times 3$ | $62 \times 256 \times 3$ | $2 \times 128 \times 3$ | $2 \times 128 \times 3$ |
| Diffusion steps | 1000 | 1000 | 1000 | 1000 | 1000 | 1000 | 1000 |
| Noise schedule | linear | linear | linear | linear | linear | linear | linear |
| Channels | 64 | 64 | 64 | 64 | 64 | 32 | 64 |
| Depth | 1 | 1 | 1 | 1 | 1 | 1 | 1 |
| Channel Multiplier | 1,2,4 | 1,2,2 | 1 | 1,2,4 | 1,2 | 1,2,4 | 1,2 |
| Attention Resolutions | 4,2,1 | 4,2,1 | 1 | 4,2,1 | 2,1 | 4,2,1 | 2,1 |
| Head Channels | 32 | 32 | 32 | 32 | 32 | 32 | 32 |
| Transformer Depth | 1 | 1 | 2 | 1 | 1 | 1 | 1 |
| Batch Size | 32 | 32 | 32 | 32 | 32 | 32 | 32 |
| Learning Rate | $8e$-5 | $8e$-5 | $5e$-4 | $5e$-4 | $8e$-5 | $5e$-4 | $5e$-4 |
| Optimizer | AdamW | AdamW | AdamW | AdamW | AdamW | AdamW | AdamW |

Table 8: Training details on different datasets.

| Dataset | B | Epochs | $\rho$ | $I_{DM}$ | Optimizer | Learning Rate | $\lambda$ | $L$ | $M$ | $M_g$ |
|---|---|---|---|---|---|---|---|---|---|---|
| Huffpost | 64 | 50 | 0.6 | 20 | Adam | $2e$-5 | 10 | 8 | 32 | 32 |
| Arxiv | 64 | 5 | 0.2 | 5 | Adam | $2e$-5 | 10 | 8 | 32 | 32 |
| Yearbook | 64 | 50 | 0.2 | 5 | Adam | $1e$-3 | 10 | 8 | 32 | 32 |
| RMNIST | 64 | 50 | 0.2 | 5 | Adam | $1e$-3 | 10 | 8 | 32 | 32 |
| fMoW | 64 | 25 | 0.6 | 30 | Adam | $2e$-4 | 10 | 8 | 32 | 32 |
| 2-Moons | 64 | 150 | 0.2 | 10 | Adam | $1e$-3 | 10 | 8 | 32 | 32 |
| ONP | 64 | 50 | 0.2 | 10 | Adam | $1e$-4 | 10 | 8 | 32 | 32 |

### D.2 Network Details

For the backbone of the task model, Huffpost and Arxiv apply pretrained DistilBERT base model [33] along with a bottleneck layer [20] to reduce the feature dimensions into 128. The bottleneck layer is implemented as the combination of a linaer layer, BatchNorm and ReLU. Yearbook uses the 4-layer convolutional network in [48], RMNIST adopts the ConvNet in [29], and fMoW employs the ImageNet-pretrained DenseNet-121 [15] along with a bottleneck layer [20] to reduce the feature dimensions into 256. Meanwhile, 2-Moons uses a MLP with two hidden layers of hidden size 64 and 128, and ONP adopts a MLP with one hidden layer of hidden size 128.

For the conditional diffusion model, we implement it in a U-Net architecture similar to LDM [31] and make some modifications to better suit our method. Detailed modifications can be found in the code provided in the supplementary material. In Table 7, we provide detailed configurations for U-Net $\mathcal{E}_{\boldsymbol{\theta}}$ on different datasets. Please refer to original paper [31] for the meaning of different hyperparameters.

### D.3 Training Recipe

Training details on different datasets are given in Table 8, where $B$ is the batch size for the task model, $I_{DM}$ is the inner iterations for updating $\mathcal{E}_{\boldsymbol{\theta}}$, $\lambda$ is the loss tradeoff hyperparameter, $\rho$ is the warm-up hyperparameter, $L$ is the maximum length of the reference point queue $Q_r$, $M$ is the maximum length of the anchor point queue $Q_a$ and prototype queue $Q_p$, and $M_g$ is the number of generated residual classifier weights based on each reference point. All experiments are conducted using the PyTorch packages and run on a single NVIDIA GeForce RTX 4090 GPU with 24GB memory.

## E More Results

### E.1 Hyperparameter Sensitivity

In Fig. 4(a), we test the sensitivity of W-Diff to the loss tradeoff hyperparameter $\lambda$, the maximum length $L$ of the reference point queue $Q_r$ and the number $M_g$ of generated residual classifier weights based on each reference point, where $\lambda \in \{0.1, 0.5, 1.0, 5.0, 10.0, 50.0\}$, $L \in \{1, 2, 4, 8\}$, $M_g \in \{8, 16, 32, 64, 128\}$. We find that larger $M_g$ results in more weights for ensemble and seems to be better. W-Diff is a little bit sensitive to $\lambda$ and $L$. Empirically, $\lambda = 10.0$ and larger $L$ work well.

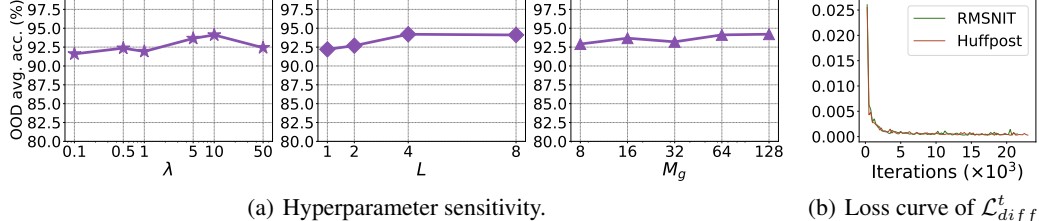

(a) Hyperparameter sensitivity.

(b) Loss curve of $\mathcal{L}_{diff}^t$

Figure 4: (a): Sensitivity of W-Diff to hyperparameters $\lambda, L, M_g$ on RMNIST. (b): The loss curve of $\mathcal{L}_{diff}^t$ on RMNIST and Huffpost when training conditional diffusion model on the second domain.

Table 9: Memory cost and inference time of diffusion model on different datasets.

| Yearbook | RMNIST | fMoW | Huffpost | Arxiv | 2-Moons | ONP |
|---|---|---|---|---|---|---|
| Number of parameters (MB) for the conditional diffusion model $\mathcal{E}_{\boldsymbol{\theta}}$ | | | | | | |
| 2.31 | 41.62 | 27.23 | 41.62 | 20.01 | 10.52 | 15.51 |
| Time (s) for generating $M_g = 32$ residual classifier weights in a batch, where denoising step $S = 1000$ | | | | | | |
| 12 | 23 | 71 | 24 | 181 | 21 | 16 |

## E.2 Convergence of Diffusion Model Training

In Fig. 4(b), we plot the loss curve of $\mathcal{L}_{diff}^t$, when the conditional diffusion model is incrementally trained on the second source domain. Form the results, we see that the noise estimation error loss $\mathcal{L}_{diff}^t$ steadily decreases and finally converges, demonstrating that using the FIFO queue to cache the recent $M$ classifier weights and prototype matrices after the warm-up stage is feasible for the diffusion model training. Storing all checkpoints after the warm-up stage provides lost of training data for diffusion model but requires more storage cost. By contrast, using a FIFO queue with a fixed length balances the storage cost and the diversity of training data.

## E.3 Memory and Time Cost of Diffusion Model

In Table 9, we list the model size and inference time of the conditional diffusion model on different datasets. Since the diffusion model only models the evolving pattern of the classifier weights which are the parameters of a linear layer, the model size that is measured by the number of parameters is small on all datasets. Besides, the inference time when forwarding the diffusion model for 1000 times to generate a batch of residual classifier weights is moderate. Concretely, forwarding the condition diffusion model for one time requires $\leq 181$ ms. Certainly, acceleration techniques, e.g., DDIM [35] can be used to further reduce the inference time.

## E.4 Significance test (t-test) of W-Diff.

To comprehensively evaluate the effectiveness of W-Diff, we conduct the significance test (t-test) on Huffpost, Arxiv, Yearbook, RMNIST and fMoW datasets. Concretely, a significance level of 0.05 is applied. If the p-value is less than 0.05, then the accuracy difference between EvoS [45] and W-Diff is statistically significant. For clearer explanation, the -log(p) of each p-value is plotted. In Fig. 5, the majority of the -log(p) of the performance comparison between EvoS [45] and W-Diff are larger than -log(0.05), which means that W-Diff is statistically superior to EvoS [45] at most datasets.

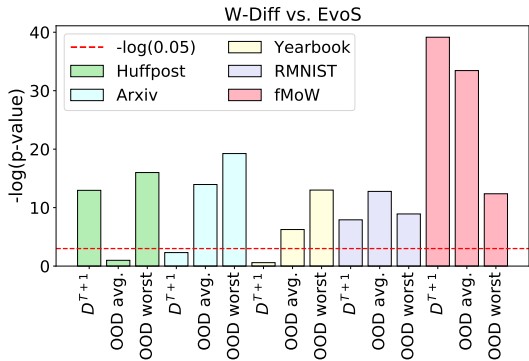

Figure 5: t-test for W-Diff *vs* EvoS [45], where a significance level of 0.05 is adopted.

