# OpenReview forum: "Weight Diffusion for Future: Learn to Generalize in Non-Stationary Environments"
_NeurIPS.cc/2024/Conference — NeurIPS 2024 poster_

### Official Review · Reviewer_BuMk · 2024-07-10

**Soundness:** 3
**Presentation:** 3
**Contribution:** 2
**Rating:** 5
**Confidence:** 4

**Summary:**

This manuscript aims to tackle the evolving domain generalization (EDG) issue, namely the domain gradually evolves in an underlying continuous structure. The paper introduces the idea of Weight Diffusion (W-Diff), a conditional diffusion model in the parameter space to learn the evolving pattern of classifiers. Combining such types of classifier with weight ensembling and a domain-shared feature space allows robust prediction. The effectiveness of the proposed method is examined on two text classification datasets, three image classification datasets, and two multi-variate classification datasets.

**Strengths:**

* The evolving domain generalization is interesting yet important to the research community.
* The manuscript is well-structured. It explains its methodology design clearly and intuitively.
* The manuscript introduces the idea of model weight generation through the diffusion model to the area of evolving domain generalization. The concept itself is interesting. However, the authors still need to discuss the related work carefully.
* The proposed method is examined on two text classification datasets, three image classification datasets, and two multi-variate classification datasets.

**Weaknesses:**

* Unclear novelty. The idea of using a diffusion model to generate model weights/heads appeared in [1, 2]. However, the manuscript did not cite and discuss these two papers (and maybe their follow-up works), making the exact contribution unclear. From the reviewer's perspective, idea 1 of using the conditional diffusion model to model parameter evolution patterns is interesting and intuitive; idea 2 of learning domain-shared feature encoder is standard. Combining idea 1 and idea 2 and applying it to the area of evolving domain generalization is ok, but still needs a careful discussion and ablation study.
* The performance gain over different datasets looks marginal.
* The manuscript needs more ablation studies. E.g., What if instead of generating the classifiers on the fly for the unseen domain, we just leverage the past classifiers for the ensemble prediction, or test-time adaptive classifier ensembling as in [3]?


### Reference
[1] Learning to Learn with Generative Models of Neural Network Checkpoints. https://arxiv.org/abs/2209.12892

[2] Diffusion-based Neural Network Weights Generation. https://arxiv.org/abs/2402.18153

[3] Adaptive Test-Time Personalization for Federated Learning. https://arxiv.org/abs/2310.18816

**Questions:**

NA

---

> ### Author Rebuttal · Authors · 2024-08-07
>
> Sincerely thanks for your efforts in reviewing the paper. Below, we respond to your questions in detail.
>
> > **Q1: Discussion with the two related papers [1, 2].**
>
> Thanks. Our method differs from [1, 2] as follows:
>
> **Focused problem**: G.pt [1] focuses on supervised learning and reinforcement learning, while our W-Diff addresses the evolving domain generalization (EDG) in the domain-incremental setting. In EDG, distribution shifts often hinder models that are trained with full supervision on in-distribution (ID) data from generalizing to out-of-distribution data. Hence, our goal is to address the distribution shift, instead of generating diverse high-performance parameters for ID data.
>
> D2NWG [2] focuses on transfer learning to provide better model parameter initialization for faster fine-tuning convergence on new datasets. However, target domains are unlabeled in EDG, preventing supervised fine-tuning. This makes D2NWG unsuitable for domain generalization. In contrast, our method generates model parameters **applicable directly to the unlabeled target domain without fine-tuning**.
>
> **Condition design for diffusion model**: G.pt [1] collects the loss/error/return of task model checkpoints during training as the condition for the diffusion model. It is designed for a single dataset to which the training data belongs, thus struggling with distribution shifts.
>
> D2NWG [2] uses CLIP to extract features for each sample and Set Transformer to generate dataset encoding from these features. The dataset encoding is used as the condition for diffusion model, while training set samples of a new dataset are required to obtain the dataset encoding. This is infeasible in unlabeled target domains.
>
> Different from [1, 2], we use the classifier weights of historical domain (referred as reference point) and the prototypes of current domain as the condition of diffusion model to generate the difference of classifier weights between reference point and anchor point (i.e., the classifier weights of current domain). Considerations of such design are 1) the difference between reference point and anchor point denotes the evolution of parameters from historical to current domain, which **conduces to modeling the crucial evolving pattern across domains in EDG**; 2) the reference point provides initialization-like information, while current prototypes offer some information about the desired decision boundary, **helping to explore the relationship between generated parameters and the given domain**.
>
> Moreover, we compare W-Diff with G.pt [1] in the following datasets. G.pt shows worse generalization on target domain, due to the limitations discussed above.
>
> ----Error rate comparison (%)---
> |Method|2-Moons|ONP|
> |-|-|-|
> |G.pt|4.5$\pm$1.2|35.1$\pm$0.8|
> |**W-Diff**|**1.5$\pm$1.0**|**32.9$\pm$0.5**|
>
> > **Q2: The idea is ok, but still needs a careful discussion and ablation study.**
>
> Thanks. We have provided the results of ablation study in **Table 4** of the PDF file. Concretely, we explore the following variants:
>
> * Variant A ablates the consistency loss $\mathcal{L}^t_{con}$ for learning domain-shared feature encoder.
>
> The performance drop of Variant A suggests that learning domain-invariant feature representations is necessary for EDG in the domain-incremental setting. Otherwise, the feature encoder could easily overfit to current domain, prohibiting task model from generalization.
>
> * Variant B ablates the conditional diffusion model and directly uses the incrementally trained classifier for inference.
>
> * Variant C directly uses the historical classifier weights in the reference point queue $Q_r$ to construct the average weight ensemble $\bar{\mathbf{W}}^{test}$ for inference: $\bar{\mathbf{W}}^{test}=\frac{1}{|Q_r|}\sum_{\ddot{\mathbf{W}}^{t^{\prime}}\in Q_r}\ddot{\mathbf{W}}^{t^{\prime}}$.
>
> The inferior results of variant B and C indicate that W-Diff benefits from generating meaningful and customized classifier weights via controlling the condition of diffusion model.
>
>
> > **Q3: The performance gain over different datasets looks marginal.**
>
> Thanks. To comprehensively evaluate the effectiveness of W-Diff, we have conducted a significance test (t-test) on different datasets. Concretely, a significance level of 0.05 is applied, and if p-value is less than 0.05, the accuracy difference between EvoS [8] and W-Diff is statistically significant. For clearer explanation, the -log(p) of each p-value has shown in red line. In **Fig. 1(b)** of the PDF file, majority of the -log(p) of the performance comparison between EvoS and W-Diff are larger than -log(0.05), which means W-Diff is statistically superior to EvoS at most datasets.
>
>
> Besides, we also extend W-Diff to regression tasks, where the regression datasets and results are from DRAIN [4]. In **Table 3** of the PDF file, W-Diff still works well. Overall, given the generalization performance on regression and text/image/multi-variate classification datasets, W-Diff is of more versatility.
>
> > **Q4: More ablation studies, e.g., just leveraging the past classifiers for the ensemble prediction, or test-time adaptive classifier ensemble as in [3]?**
>
> Thanks. As you suggested, we have added more ablation study results in **Table 1** of the PDF file.
>
> * Variant C in the reply to **Q2** denotes leveraging the past classifiers for the ensemble prediction.
>
> * Variant D denotes using the APT-batch manner in [3], which first conducts unsupervised adaptation of classifier by back-propagating the gradient of entropy loss to source-trained classifier and then makes independent predictions on each batch.
>
> * Variant E denotes using the APT-online manner in [3], which first updates source-trained classifier with the cumulative moving average of update directions from the first target data batch to current batch, and then makes predictions on current batch.
>
> In **Table 1** of the PDF file, W-Diff outperforms these variants, because they ignore the evolving pattern which is crucial for EDG.

---

> > ### Comment · Reviewer_BuMk · 2024-08-08
> > **Thank you for the rebuttal**
> >
> > The reviewer thanks for the detailed responses and additional results. The reviewer decided to raise the score to 5.

---

> > > ### Author Response · Authors · 2024-08-08
> > > **Thanks for upgrading score**
> > >
> > > We really appreciate your feedback and are pleased that our rebuttal has addressed your concerns, leading to a raised rating! In future revisions, we will update the manuscript according to your suggestions.

---

### Official Review · Reviewer_Ve2z · 2024-07-12

**Soundness:** 3
**Presentation:** 3
**Contribution:** 3
**Rating:** 7
**Confidence:** 3

**Summary:**

The paper proposes a novel method called Weight Diffusion (W-Diff), which employs a conditional diffusion model in the parameter space to learn the evolving patterns of classifiers during domain-incremental training. During inference, the proposed method uses an ensemble of classifiers tailored to the target domain for robust predictions.

**Strengths:**

1. This work is pioneering in applying diffusion models for generating parameters in a practical context.
2. The paper is well-written with detailed descriptions of each component of the algorithm.
3. The paper demonstrates the effectiveness of W-Diff through comprehensive experiments on both synthetic and real-world datasets.
4. The use of an ensemble of classifiers enhances prediction robustness.

**Weaknesses:**

1. The experiments are conducted on relatively small networks, and the diffusion model is used solely to generate the classifier. It remains unclear whether this method can scale to larger networks.
2. The paper should include a comparison with Variational Autoencoders (VAEs) or other generative models. Given that diffusion models are more complex and challenging to train, it is important to demonstrate the necessity of using them.
3. The paper should include a comparison of training time with state-of-the-art algorithms.
4. It would be good to include a notation table in the appendix, as there are numerous notations used throughout the paper.

**Questions:**

See Weaknesses.

**Limitations:**

The limitation of the paper is that it only considers generating the classifier. It would be interesting to explore whether it can generate the entire network.

---

> ### Author Rebuttal · Authors · 2024-08-07
>
> Thanks for your efforts in reviewing the paper and the constructive comments. Below, we have tried our best to address your concerns in detail.
>
> > **Q1: Can this method scale to larger networks and generate the entire network?**
>
> Thanks. Firstly, we have tried larger networks on the fMoW dataset by replacing the DenseNet-121 with DenseNet-161, DenseNet-169, DenseNet-201, respectively. The results are provided in **Table 2** of the PDF file, where further performance improvements are obtained.
>
> Secondly, our method can be extended to generate more parameters, but some modifications are needed to solve the different shapes of parameters from multiple layers and the training efficiency of diffusion models when the number of parameters to be generated is high. One possible solution to the shape difference is to flatten parameters from different layers and then concatenate all the flattened parameters into a vector. The generated parameters in a vector format are finally converted into their original shapes. As for the training efficiency, we can additionally train a VAE to encode the high-dimensional parameter vector into a low-dimensional embedding and then decode the embedding to reconstruct the parameter vector. Later, the diffusion model is trained in the low-dimensional embedding space and the decoder of VAE is used to recover parameters from the generated embedding via diffusion model in the inference stage. As stated in the Limitations section, we will leave this for a future work due to time constraints.
>
> Thirdly, as mentioned above, generating the entire network is possible, but it is not a good choice for evolving domain generalization (EDG).
>
> 1) The massive parameters of the entire network require a large number of sequential source domains to accurately model the evolving pattern of the entire network.
>
> 2) Not all parameters are unshared. Shallow layers of the network usually extract general knowledge that can be shared, while deep layers are specific to tasks or datasets [5].
>
> 3) Previous EDG work [7] has provided theoretical evidence, showing that solely learning dynamic features is insufficient. [6] and [7] both use a static variational encoding network as the feature encoder to extract invariant features and train a domain-adaptive classifier. Hence, based on the experience and conclusion of previous works, we choose to only generate the parameters of task head, e.g., the classifier for classification tasks.
>
> 4) Moreover, in the paper, we compare our W-Diff with DRAIN [4] which leverages LSTM to learn the evolving pattern of the entire network. Experimentally, on 2-Moons and ONP datasets, the error rate of target domain is **3.2% (DRAIN) vs. 1.5% (W-Diff), 38.3% (DRAIN) vs. 32.9% (W-Diff)**. These results partially show that generating entire network is less effective for EDG.
>
>
> > **Q2: Comparison with VAEs.**
>
> Thanks for your advice. We have added the results when replacing the diffusion model with VAE to generate classifier weights, conditioned on the reference point and prototypes. Please refer to **variant F** in **Table 4** of the PDF file. The better generalization performance when using diffusion model to learn the evolving pattern of classifiers shows its superiority in modeling complex distributions and generating high-quality data, which has also been proven by many diffusion-based generation works. Besides, to reduce the difficulty of training diffusion models, we only generate the parameters of the classifier. The number of parameters (MB) for the conditional diffusion model is given in **Table 6** of the PDF file, from which we see that the diffusion model is small and easy to train.
>
>
> > **Q3: Comparison of training time with SOTA method.**
>
> Thanks. The training time complexity of our method mainly comes from the diffusion model. For simplicity, we take the U-Net with all convolutional layers as an example. Assuming that there are $L$ convolutional layers, the size of feature map in the $i$-th layer is $H_i \times W_i×C_i$ and the kernel size is $k_i\times k_i$. Then the time complexity of one forward pass is $\mathcal{O}(\sum_{i=1}^L H_i \times W_i \times C_i \times C_{i-1} \times k_i^2)$. Let $S$ denote the total time step in diffusion model. Then the time complexity of training diffusion models for $I$ iterations can be approximated as $\mathcal{O}(I \times S \times \sum_{i=1}^L H_i\times W_i \times C_i\times C_{i-1}\times k_i^2)$.
>
> Moreover, in **Table 5** of the PDF file, we compare the training time and GPU memory of our W-Diff, DRAIN [4], EvoS [8] and GI [11] on the RMNIST and Huffpost datasets. We acknowledge that our method has no significant advantage in terms of training time, due to the diffusion model. But this is not a limitation unique to our approach and most methods based on diffusion models have this limitation. As part of future work, we will investigate and try to address the limitation to further enhance the training efficiency.
>
> In addition, it is worth mentioning that GI and DRAIN require huge computational resources during the training process, when they are applied on relatively large networks. Specifically, DRAIN needs to generate the entire network parameters and the fine-tuning stage of GI requires second-order gradients. On the Huffpost dataset with the backbone of DistilBERT-base and a batch size of 64, GI and DRAIN encounter the issue of **GPU memory explosion**. By contrast, our method utilizes diffusion model to generate only classifier weights, and as shown in **Table 6** of the PDF file, the diffusion model is small to train, without the explosion of GPU memory when using the same batch size of 64. Overall, the main contribution of our work is providing a new perspective to address EDG in the domain-incremental setting via delicately tailoring the conditional diffusion model.
>
>
> > **Q4: Include a notation table in the appendix.**
>
> Thanks for your advice. We will add a notation table in the revision to ease the burden of readers.

---

> > ### Comment · Reviewer_Ve2z · 2024-08-08
> >
> > Thank you very much for your detailed reply. My concerns are addressed and I will keep my score towards acceptance.

---

> > > ### Author Response · Authors · 2024-08-10
> > > **Thanks for positive feedback**
> > >
> > > Thank you so much for your positive feedback and dedicated time to review our paper! We are glad to know that our rebuttal and new experiments have addressed your concerns.

---

### Official Review · Reviewer_tBNz · 2024-07-13

**Soundness:** 3
**Presentation:** 2
**Contribution:** 2
**Rating:** 5
**Confidence:** 3

**Summary:**

This paper presents Weight Diffusion (W-Diff), a framework for domain generalization in non-stationary environments. W-Diff leverages a conditional diffusion model in the parameter space to learn the evolving pattern of classifiers during domain-incremental training. Experiments on synthetic and real-world datasets demonstrate its superior generalization performance on unseen future domains

**Strengths:**

**S1.**
This paper introduces a novel approach for capturing evolving patterns at the parameter level using diffusion models.

**S2.**
The proposed method demonstrates good performance in generalizing to unseen domains on diverse datasets.

**S3.**
This paper addresses the practical challenge of sequentially arriving non-static source domains, mimicking real-world scenarios.

**Weaknesses:**

**W1.**
The paper does not sufficiently justify why a diffusion model is chosen for the domain generalization (DG) problem. While diffusion models have shown excellent performance in generation tasks, the specific advantages they offer for DG are not clearly articulated.

**W2.**
The computational complexity of the diffusion model training might be a barrier for very large datasets or real-time applications. A thorough analyses on the computational complexity espeically for training is needed.

**W3.**
The evaluation is limited to classification tasks; applicability to other types of tasks, such as regression, remains unexplored.

**Questions:**

Please refer to my weaknesses section.

**Limitations:**

The authors discussed some limitations of this work in their paper.

---

> ### Author Rebuttal · Authors · 2024-08-07
>
> We are grateful for your efforts in reviewing the paper as well as your constructive comments. Below, we do our utmost to address your concerns.
>
> > **Q1: The specific advantages that diffusion models offer for DG.**
>
> Thanks for your comment. Firstly, instead of learning a deterministic classifier like previous EDG methods [4, 8], we consider the classifier as a distribution, which conduces to improving the robustness and reducing miscalibration. Yet, the prior knowledge of distribution type is unknown and we only have the observed data (i.e., saved classifier checkpoints) from the unknown distribution. Fortunately, diffusion models are quite powerful in modeling complicated and unknown distribution based on observed data. Moreover, we have also tried other generative models, e.g., VAE, to generate the classifier weights. The results on RMNIST dataset are shown in the following table. We see that the generalization performance is worse when using VAE, which demonstrates the superiority of diffusion model in modeling complicated classifier distributions.
>
> -----Accuracy (%) on RMNIST (K = 3)-----
> |method|generative model|$\mathcal{D}^{T+1}$|OOD avg.|OOD worst|
> |:---:|:---:|:---:|:---:|:---:|
> |W-Diff|VAE| 98.66|93.83|87.16|
> |W-Diff|diffusion model|**98.70**|**94.12**|**87.36**|
>
> Secondly, in the paper, we focus on the problem of evolving domain generalization (EDG), where the domain gradually evolves over time in an underlying pattern. In addition to the invariant feature learning, it is also critical to excavate the underlying evolving pattern across domains for better predicting the model status on future domains. Inspired by the fact that conditional diffusion models excel at generating specific images using additional information, we delicately design the condition including both the classifier weights of historical domain and the prototypes of current domain, and train the conditional diffusion model to generate the discrepancy of classifier weights between historical domain and current domain. The discrepancy represents the evolving of classifier across domains. In such way, we convey the information on classifier evolving from past to present to the conditional diffusion model.
>
> Thirdly, during inference, we can inject the target data information through the prototypes in the condition to generate target-customized classifiers. Besides, since the diffusion model is stochastic in nature, multiple noise samplings can generate different classifiers, forming an ensemble, which leads to better robustness for generalizing on new domains.
>
> > **Q2: The computational complexity of the diffusion model training.**
>
> Thanks for your advice. For simplicity, we take the U-Net with all convolutional layers as an example. Assuming that there are $L$ convolutional layers, the size of feature map in the $i$-th layer is $H_i \times W_i×C_i$ and the kernel size is $k_i\times k_i$. Then the time complexity of one forward pass is $\mathcal{O}(\sum_{i=1}^L H_i \times W_i \times C_i \times C_{i-1} \times k_i^2)$. Let $S$ denote the total time step in diffusion model. Then the time complexity of training diffusion models for $I$ iterations can be approximated as $\mathcal{O}(I \times S \times \sum_{i=1}^L H_i \times W_i \times C_i \times C_{i-1} \times k_i^2)$.
>
> Moreover, in **Table 5** of the PDF file, we compare the training time and GPU memory of our W-Diff, DRAIN [4], EvoS [8] and GI [11] on the RMNIST and Huffpost datasets. We acknowledge that our method has no significant advantage in terms of training time, due to the diffusion model. But this is not a limitation unique to our approach and most methods based on diffusion models have this limitation. As part of future work, we will investigate and try to address the limitation to further enhance the training efficiency.
>
> In addition, it is worth mentioning that GI and DRAIN require huge computational resources during the training process, when they are applied on relatively large networks. Specifically, DRAIN needs to generate the entire network parameters and the fine-tuning stage of GI requires second-order gradients. On the Huffpost dataset with the backbone of DistilBERT-base and a batch size of 64, GI and DRAIN encounter the issue of **GPU memory explosion**. By contrast, our method utilizes diffusion model to generate only classifier weights, and as shown in the following table, the diffusion model is small to train, without the explosion of GPU memory when using the same batch size of 64. Overall, the main contribution of our work is providing a new perspective to address EDG in the domain-incremental setting via delicately tailoring the conditional diffusion model.
>
> -----------------Number of parameters (MB) of conditional diffusion model $\mathcal{E}_{\boldsymbol{\theta}}$----------------
> |Yearbook|RMNIST|fMoW|Huffpost|Arxiv|2-Moons|ONP|
> |:---:|:---:|:---:|:---:|:---:|:---:|:---:|
> |2.31|41.6|27.23|41.62|20.01|10.52|15.51|
>
>
> > **Q3: The evaluation is limited to classification tasks; applicability to other types of tasks, such as regression, remains unexplored.**
>
> Thanks for your advice. We have extended our method to the two regression datasets (i.e., House and Appliance) in DRAIN [4]. For regression tasks, the prototype is calculated at the domain level, i.e., the average of features in a single domain. And the cross-entropy loss is replaced with the mean squared error (MSE) loss and the Kullback-Leibler divergence in the consistency loss is replaced with MSE. Specifically, the results are provided in the following table, where our method still achieves better generalization performance.
>
> -----------------Mean absolute error (MAE) for regression tasks----------------
> | Method | House | Appliance |
> |:---: | :---: | :---:|
> |Offline |11.0$\pm$0.36|10.2$\pm$1.1|
> |IncFinetune |9.7$\pm$0.01|8.9$\pm$0.5|
> |CIDA |9.7$\pm$0.06|8.7$\pm$0.2|
> |GI |9.6$\pm$0.02|8.2$\pm$0.6 |
> |DRAIN |9.3$\pm$0.14|6.4$\pm$0.4 |
> |**W-Diff** |**9.1$\pm$0.15** |**4.9$\pm$0.3**|

---

> > ### Comment · Reviewer_tBNz · 2024-08-10
> >
> > I appreciate the author's responses, which addressed most of my previous concerns, though the computational cost associated with the use of the diffusion model remains a consideration. I will maintain my current score of 5 and take the rebuttal into account in the next phase of discussion.

---

> > > ### Author Response · Authors · 2024-08-10
> > > **Further response to concerns about the computational cost**
> > >
> > > Thanks for your feedback and we are happy to know that our rebuttal has addressed most of your concerns. As for the computational cost during the training process, we would like to provide further clarifications.
> > >
> > > **Firstly**, the problem we focus on is the evolving domain generalization (EDG) in the domain-incremental setting, which is an under-explored area, compared with previous EDG. The consideration of domain-incremental setting mimics the dynamics of training domains in the real-world, which is more practical yet challenging. The resulting benefit is that once a new training domain is available, we can incrementally train the model only on the new domain, instead of training from scratch with all old domains and the new domain using previous non-incremental EDG methods. The latter would be inefficient when new domains continually emerge with the passage of time.
> > >
> > > **Secondly**, the training time we report in previous response to **Q2** is the total training time when the number of source domains is $T$. In the following table, we further compared the average training time per domain and the performance improvement over the baseline GI. Though our W-Diff indeed requires more training time per domain, we think the increment of training time is acceptable, considering the significant performance improvement over GI. Besides, if a new source domain of similar dataset size comes, W-Diff just requires roughly the average training time to train the model on the new source domain.
> > >
> > > -------------------------------Computational cost and performance comparison on **RMNIST** dataset (T=6, K=3)-------------------------------
> > > | Method | Total training time (h) | Average training time per domain (h) | GPU memory (GB) | $\mathcal{D}^{T+1}$ accuracy (%) | OOD avg. accuracy (%) | OOD worst accuracy (%) |
> > > | :-: | :-: | :-: | :-: | :-: | :-: | :-: |
> > > | GI | 2.40 | 0.40 | 1.9 | 97.78 | 91.00 | 82.46 |
> > > | W-Diff | 4.61 | 0.77 | 4.0 | 98.70 | 94.12 | 87.36 |
> > > | Increment $\Delta$ of W-Diff over GI| 2.21 | 0.37 | 2.1 | **0.92** | **3.12** | **4.90** |
> > >
> > > -------------------------------Computational cost and performance comparison on **Huffpost** dataset (T=4, K=3)-------------------------------
> > > | Method | Total training time (h) | Average training time per domain (h)| GPU memory (GB) | $\mathcal{D}^{T+1}$ accuracy (%) | OOD avg. accuracy (%) | OOD worst accuracy (%) |
> > > | :-: | :-: | :-: | :-: | :-: | :-: | :-: |
> > > | GI$^\clubsuit$ | 10.04 | 2.51 | 17.3 | 64.96 | 63.11 | 60.15 |
> > > | W-Diff | 12.61 | 3.15 |15.6 | 73.91 | 72.29 | 70.40 |
> > > | Increment $\Delta$ of W-Diff over GI | 2.57 | 0.64 | **-1.7** | **8.95** | **9.18** | **10.25** |
> > >
> > > *(GI$^\clubsuit$ denotes that a much smaller batch size is used due to the **GPU memory explosion** when using the method on Huffpost dataset with the backbone of DistilBERT base.)*
> > >
> > > **Thirdly**, it is noteworthy that GI requires a pre-training stage on all domains and then sequentially finetunes the pre-trained model on each domain. That is, if a new source domain comes (i.e., $T$ changes), it needs to re-execute the pre-training stage from scratch. By contrast, our W-Diff incrementally trains the previously trained model on the new source domain, rather than training from the first source domain every time. Indeed, in the fixed $T$, our W-Diff is at a disadvantage in terms of training time. But in practice, **in the long run, our method is more advantageous in training time**. To verify this, we record the total training time of GI and W-Diff on RMNIST dataset, when $T$ increases in a sequence of $2, 3, 4, 5, 6$. Concretely, the result is **`6.74h (GI) vs 4.61h (W-Diff)`**. Here, the training time of W-Diff is the same as in the above table, because the training procedure of W-Diff has already simulated the increasing of $T$.
> > >
> > > Certainly, further improving the training efficiency of W-Diff is a nice direction for future work. We hope that our response could mitigate your concerns about computational cost. Looking forward to your feedback. Thanks so much!

---

### Official Review · Reviewer_EUGW · 2024-07-16

**Soundness:** 3
**Presentation:** 2
**Contribution:** 3
**Rating:** 6
**Confidence:** 4

**Summary:**

This paper deals with the problem of evolving domain generalization in non-stationary environments, where dynamically changing source domains arrive sequentially, but we only have access to data samples from the current domain (and not the past ones). The main idea is to learn a conditional diffusion model that predicts how the evolving pattern of the classifiers, as the source domains keep changing. In a nutshell, the conditional diffusion model learns how to go from past classifier weights (reference points) to the weights of the current classifier (anchor point), conditioned on prototypes of the current domain. One challenge concerns the shared feature encoder: if the encoder is updated using the latest source domain, then it will overfit to it. To avoid this, the authors suggest to enforce prediction consistency among multiple classifiers, so that all classifiers from past and present domains give the same prediction given the same feature representation. This implies that the past reference points do not become obsolete. During inference, the goal is to predict the classifier weights given the current domain prototypes. For this purpose, the framework uses the conditional diffusion model to cheaply generate a large number classifiers, which it then averages for improved robustness. Extensive experiments on synthetic and real datasets show strong performance for the proposed framework.

**Strengths:**

- Overall, the framework is interesting. The use of conditional diffusion models makes sense in thee context of non-stationary adaptation, as such models can capture the dynamically changing source domains. Furthermore, it improves robustness because it is possible to generate multiple weight predictions, and then take their average.
- The consistency constraint is a nice way to force the shared representation that remains valid for old classifiers, even as the source domain keeps changing. The use of reference points makes it possible to completely get rid of old data (except the data related to the current source domain), and is efficient. The diffusion models are able to capture even complex non-stationarities, which gives the model significant expressive power.
- The experimental study is quite extensive and shows strong performance across a variety of benchmarks for the proposed weight-diffusion framework.

**Weaknesses:**

- In inference phase, the authors assume they know the dataset (without the labels) for the future timesteps $\{T+1,\dots,T+K\}$. Of course, this cannot work when the datasets are not given. In such a case, the model must also be in a position to infer the future data distribution for the points $x_i$, in order to then estimate the prototypes $f_{test}$. The current framework seems not to be able to deal with that. I think it would help if the authors explain what the limitations of their framework are, and why for instance it is very difficult to predict the future data distribution in non-stationary environments. The current weight-diffusion model assumes that we can have access to future data but not their labels during inference, but this may not always be the case.
- I like the fact that the authors condition on the context. But I was not clear why the context only consists of the historical weights and the current prototypes. Why not also include the timestep difference between the reference point and the current timestep? For instance, assume we are given the context (reference point + prototype). If the reference point is from 5 timesteps ago, then the predicted weights might be different compared to the case where the reference point is from 10 timesteps ago, even if the reference point value is the same. If we are not given how far into the past the reference point lies, then I was thinking that predicting the current weights is harder. On the other hand, perhaps this is already taken into account implicitly, in the framework, but this is not immediately clear.
- The point above also points to the limitation of the current framework, which is a lack of formal theory. It works very well  experimentally, but some aspects are not very clear.
- There are numerous typos and bad grammar throughout the paper. The authors should do a very careful proofreading and fix all the errors.

**Questions:**

- Why is it not necessary to condition on the timestep difference between the reference point and the current anchor? Intuitively, one might expect that knowing how far into the past the reference point is could lead to better predictions. Did the authors try this?
- I was under the impression that U-Net still needs the parameters $\beta_s$ in Equations (3) and (4).. If yes, how did the authors set these hyperparameters? Where they fixed throughout training?
- I am not sure I can understand Figure 3a. $\tilde{W}^{7\mid 5}$ is concentrated in the upper left, whereas $W^7$ is at the very bottom. And, yet, both of them perform very well in Figure (3b). What does that exactly say about the computed weights? Maybe it means that the visualization does not really tell us anything about test performance?
- In the ablation study in Section 5.3, what is Variant B?
- Can the authors be more clear about the limitations of their framework? Could their framework be modified to also predict the future data distribution (e.g., how the $x_i$ will be distributed in the next timestep)?

**Limitations:**

No concern.

---

> ### Author Rebuttal · Authors · 2024-08-07
>
> Thanks a lot for your efforts in reviewing the paper. Below, we respond to your questions in detail.
>
> > **Q1: Concerns about the assumption of dataset accessibility in the inference phase.**
>
> Thanks. Firstly, we would like to make it clear that for discriminative tasks, test data (i.e., target data in our paper) must have been given, when the model is used for testing. Thus, our method doesn't need to infer future data distributions. We can estimate the prototype matrix directly by feeding test data into the model to get their features and class probability predictions.
>
> Secondly, previous evolving domain generalization (EDG) works [4, 6, 7, 11] are only evaluated on the next one target domain. Following [8, 10], we evaluate on the next $K$ target domains, hoping models can also generalize well on the target domain in the farther future. When $K=1$, it aligns with previous standard evaluations. Besides, the evaluation on each domain is conducted independently.
>
> Finally, given that the whole data of a target domain may not come at once, we also offer a way for estimating the prototype matrix in a batch-data stream scenario, where the prototype matrix for the $j$-th target data batch is estimated using the cumulative moving average. In **Fig. 1(a)** of the PDF file, we provide the results on RMNIST and Huffpost when evaluating W-Diff in the batch-data stream scenario.
>
>
> > **Q2: Why not include timestamp difference into the condition?**
>
> Thanks. In our method, timestamp difference information is implicitly considered, as domains evolve over time and arrive in chronological order during training. Besides, the difference between classifier weights of current and historical domains also contains implicit timestamp difference information. Moreover, due to different optimization starting points and stochastic gradient descent, it is unlikely that classifiers respectively trained for two domains are exactly the same. Hence, we do not explicitly include timestamp difference in the condition.
>
> Nevertheless, as you suggested, we have also tried to explicitly include the timestamp difference $\Delta t$. Results on RMNIST are shown in **Table 1** of the PDF file, where no obvious improvement is obtained, possibly due to reasons above.
>
> > **Q3: The framework works very well experimentally but lacks formal theory.**
>
> Thanks. The gradient descent algorithm (GDA) is known to be theoretically supported, and we find the denoising process of diffusion models can be seen as a learnable GDA with momentum update and small noise perturbation. Let us consider the following gradient descent algorithm for optimizing the model with parameters $\Phi$ on dataset $D$ via minimizing loss $L$:
>
> $\Phi_{i+1}=\Phi_i-\lambda \cdot \nabla_{\Phi}L(\Phi_i, D),\quad i=0,\ldots,T-1,$
>
> where $\lambda$ controls the update step size.
>
> Alternatively, we generate classifier weights via conditional diffusion model $\mathcal{E}_{\theta}$ with the following denoising process:
>
> $W^t\_{s-1}=\frac{1}{\sqrt{\alpha_s}}\left(W^t\_s - \frac{\beta\_s}{\sqrt{1-{\bar{\alpha}}\_s}} \mathcal{E}\_\theta(W^t\_s,s,\ddot W^{t^{\prime}}, \mu^t)\right) + \sigma\_s \epsilon, \quad \epsilon \sim N(\mathbf 0,\mathbf I),$
>
> where $s$ is diffusion step, $W^t_0$ is the overfitted classifier weights on the $t$-th domain $D^t$, $\mu^t$ is the prototype matrix of $D^t$, and $\ddot W^{t^{\prime}}$ is the saved overfitted classifier weights on historical domain $D^{t^{\prime}}, t^{\prime} < t$.
>
> Then, we can reformulate the denoising process as
>
> $W^t\_{s-1}=W^t\_s-\underbrace{\frac{\beta\_s}{\sqrt{\alpha\_s}\sqrt{1-{\bar{\alpha}}\_s}}}\_{\lambda} \mathcal{E}\_{\theta}(W^t\_s,s, \ddot W^{t^{\prime}}, \mu^t)+\underbrace{(\frac{1}{\sqrt{\alpha\_s}}-1)W^t\_s}\_{\text{momentum update}}+\underbrace{\sigma\_s \epsilon}\_{\text{noise perturbation}}.$
>
> Compared with GDA, the conditional diffusion model can be viewed as learning a descent path of $\nabla_W L(W^t_s, D^t|\ddot W^{t^{\prime}}, \mu^t)$, so that conditioned on historical classifier weights and current prototype matrix, the diffusion model can directly generate classifier weights suitable for current domain.
>
> > **Q4: Typos and grammar errors.**
>
> Thanks. We have proofread the paper carefully and corrected the errors.
>
> > **Q5: Settings of $\beta_s$ in U-Net.**
>
> Thanks. Eq.(3) is obtained by using $\alpha_s=1-\beta_s$. And $\beta_s$ is set via the following code, and betas is fixed throughout training.
>
> ```
> betas=(torch.linspace(1e-4 ** 0.5, 2e-2 ** 0.5, 1000) ** 2)
> ```
>
> > **Q6: Explanation of Figure 3a.**
>
> Thanks. What we want to convey is that our generated classifier weights are diverse and generally high-performance. $W^7$ is obtained by fine-tuning the classifier on domain $\mathcal{D}^7$, which may be trapped in local optima. The parameter space is too big to fully explore during fine-tuning. It does not mean that only the area around $W^7$ is good. $\hat W^{7|5}$ generally performs well, meaning that our generated classifier weights are diverse and potentially cover better but unexplored area in the fine-tuning process.
>
>
> > **Q7: What is Variant B in ablation study?**
>
> Thanks. Variant B ablates the conditional diffusion model and directly uses the incrementally trained classifier along with the learned domain-shared feature encoder for inference.
>
> > **Q8: Could it be modified to predict future data distribution?**
>
> Thanks. Predicting future data distribution is possible by saving partial historical data to learn the evolving pattern at the data level, but it is not wise for discriminative tasks (e.g., classification), which requires further fine-tuning on generated instances to generalize discriminative models to target domain. Moreover, due to varied data types (e.g., image, text, multivariate in our experiments), data diffusion is cumbersome, which requires quite different architectures and lacks of universality. By contrast, our weight diffusion is more general for discriminative tasks on different datatypes.

---

> > ### Comment · Reviewer_EUGW · 2024-08-10
> > **thank you for response**
> >
> > I thank the authors for their rebuttal. I will keep my original score for now (which anyway leans towards acceptance), but may revise my score upward in the next phase.

---

> > > ### Author Response · Authors · 2024-08-10
> > > **Thanks for positive feedback**
> > >
> > > We sincerely appreciate your valuable reviews and positive feedback. We hope the idea and approach presented in this work can inspire more studies in this direction.

---

### Author Rebuttal · Authors · 2024-08-07

Sincerely thank all the reviewers for their efforts in reviewing our paper and providing constructive suggestions. We are greatly encouraged that the reviewers find that

* our framework/idea is **interesting** (*Reviewer EUGW and BuMk*), **novel** (*Reviewer tBNz*), and **pioneering** in applying diffusion models for generating parameters in a **practical** context (*Reviewer Ve2z*);
* the considered problem of evolving domain generalization is **practical** (*Reviewer tBNz*), **interesting yet important** (*Reviewer BuMk*);
* our paper is **well-written** with detailed descriptions (*Reviewer Ve2z*) and **well-structured** (*Reviewer BuMk*);
* the experimental study is **extensive** (*Reviewer EUGW*) and **comprehensive** (*Reviewer Ve2z*);
* and the performance is **strong** across a variety of benchmarks (*Reviewer EUGW*), **good** in generalizing to unseen domains on **diverse** datasets (*Reviewer tBNz*).

As for the concerns and suggestions raised by each reviewer, we have done our best to address them thoroughly and have provided detailed responses to each of them. Below are references that we used in our replies to each reviewer. Additionally, a `one-page PDF` file that includes the relevant figures and tables referenced in our replies has been uploaded. Please refer to this PDF file for detailed results, if needed.

Refs:

[1] Learning to Learn with Generative Models of Neural Network Checkpoints. arXiv:2209.12892, 2022.

[2] Diffusion-based Neural Network Weights Generation. arXiv:2402.18153, 2024.

[3] Adaptive Test-Time Personalization for Federated Learning. In NeurIPS, 2023.

[4] Temporal domain generalization with drift-aware dynamic neural networks. In ICLR, 2023.

[5] How transferable are features in deep neural networks? In NeurIPS 2014.

[6] Generalizing to evolving domains with latent structure-aware sequential autoencoder. In ICML, 2022.

[7] Enhancing Evolving Domain Generalization through Dynamic Latent Representations. In AAAI, 2024.

[8] Evolving standardization for continual domain generalization over temporal drift. In NeurIPS, 2023.

[9] High-resolution image synthesis with latent diffusion models. In CVPR, 2022.

[10] Wild-time: A benchmark of in-the-wild distribution shift over time. In NeurIPS, 2022.

[11] Training for the future: A simple gradient interpolation loss to generalize along time. In NeurIPS, 2021.

---

### Comment · Area_Chair_DHGp · 2024-08-09
**Discussion period starts**

Dear reviewers,

Thank you for your valuable contributions to the NeurIPS review process! The author-reviewer discussion period has now begun. I’ve noticed that the ratings for this paper are a little dispersed, and opinions among the reviewers are not fully aligned. This makes our discussion even more crucial to reach a consensus and ensure a fair and thorough evaluation. Please engage actively with the authors during this period. If you have any questions or need further clarification on any points, this is the best time to address them directly with the authors.

best,

AC

---

### Decision · Program_Chairs · 2024-09-25

**Decision:**

Accept (poster)

**Comment:**

**Summary**

This paper proposes a novel method called Weight Diffusion (W-DiFF) for EDG. The approach leverages a conditional diffusion model to learn the evolving patterns of classifiers during domain-incremental training, allowing the model to generalize more effectively to unseen future domains. The method is validated through extensive experiments across various datasets, including text, image, and multi-variate classification tasks.


**Decision**

The decision to accept is based on the proposed method’s ability to tackle a challenging and practical setting of EDG, where source domains are not simultaneously available. This represents an advancement over existing approaches that typically assume all source domains are available concurrently.  Additionally, the method introduces an innovative application of a conditional diffusion model to learn evolving patterns in classifier weights, which, while subject to some debate among the reviewers, adds a novel perspective to the field of domain generalization. The strong performance across multiple benchmarks, combined with the comprehensive rebuttal that addressed concerns (e.g., computational cost) and provided further validation, supports the acceptance of this paper.

For the final version, please ensure that the additional experiments and comparisons provided during the rebuttal are included, as they significantly strengthen the paper's contribution.